# ConsistI2V: Enhancing Visual Consistency for Image-to-Video Generation

**Weiming Ren**[1 2 3]**, Huan Yang**[3]**, Ge Zhang**[1 3]**, Cong Wei**[1 2]**, Xinrun Du**[3]**, Wenhao Huang**[3]**,
Wenhu Chen**[1 2]

[1]**University of Waterloo,** [2]**Vector Institute,** [3]**01.AI**
`{w2ren,wenhuchen}@uwaterloo.ca, hyang@fastmail.com`

Reviewed on OpenReview: `https://openreview.net/forum?id=vqniLmUDvj`

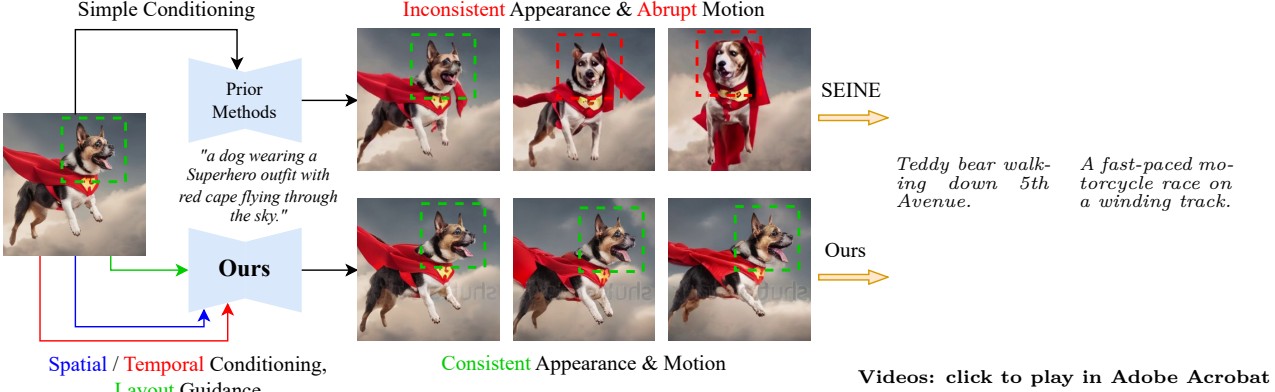

Figure 1: Comparison of image-to-video generation results obtained from SEINE (Chen et al., 2023c) and our ConsistI2V. SEINE shows degenerated appearance and motion as the video progresses, while our result maintains visual consistency. We feed the same first frame to SEINE and ConsistI2V and show the generated videos on the right.

## Abstract

Image-to-video (I2V) generation aims to use the initial frame (alongside a text prompt) to create a video sequence. A grand challenge in I2V generation is to maintain visual consistency throughout the video: existing methods often struggle to preserve the integrity of the subject, background, and style from the first frame, as well as ensure a fluid and logical progression within the video narrative (*cf.* Figure 1). To mitigate these issues, we propose ConsistI2V[1], a diffusion-based method to enhance visual consistency for I2V generation. Specifically, we introduce (1) spatiotemporal attention over the first frame to maintain spatial and motion consistency, (2) noise initialization from the low-frequency band of the first frame to enhance layout consistency. These two approaches enable ConsistI2V to generate highly consistent videos. We also extend the proposed approaches to show their potential to improve consistency in auto-regressive long video generation and camera motion control. To verify the effectiveness of our method, we propose I2V-Bench, a comprehensive evaluation benchmark for I2V generation. Our automatic and human evaluation results demonstrate the superiority of ConsistI2V over existing methods.

---

[1]Project Website: `https://tiger-ai-lab.github.io/ConsistI2V/`

# 1 Introduction

Recent advancements in video diffusion models (Ho et al., 2022b) have led to an unprecedented development in text-to-video (T2V) generation (Ho et al., 2022a; Blattmann et al., 2023). However, such conditional generation techniques fall short of achieving precise control over the generated video content. For instance, given an input text prompt "a dog running in the backyard", the generated videos may vary from outputting different dog breeds, different camera viewing angles, as well as different background objects. As a result, users may need to carefully modify the text prompt to add more descriptive adjectives, or repetitively generate several videos to achieve the desired outcome.

To mitigate this issue, prior efforts have been focused on encoding customized subjects into video generation models with few-shot finetuning (Molad et al., 2023) or replacing video generation backbones with personalized image generation model (Guo et al., 2023). Recently, incorporating additional first frame images into the video generation process has become a new solution to controllable video generation. This method, often known as image-to-video generation (I2V)[2] or image animation, enables the foreground/background contents in the generated video to be conditioned on the objects as reflected in the first frame.

Nevertheless, training such conditional video generation models is a non-trivial task and existing methods often encounter appearance and motion inconsistency in the generated videos (shown in Figure 1). Initial efforts such as VideoComposer (Wang et al., 2023c) and VideoCrafter1 (Chen et al., 2023a) condition the video generation model with the semantic embedding (e.g. CLIP embedding) from the first frame but cannot fully preserve the local details in the generated video. Subsequent works either employ a simple conditioning scheme by directly concatenating the first frame latent features with the input noise (Girdhar et al., 2023; Chen et al., 2023c; Zeng et al., 2023; Dai et al., 2023) or combine the two aforementioned design choices together (Xing et al., 2023; Zhang et al., 2023a) to enhance the first frame conditioning. Despite improving visual appearance alignment with the first frame, these methods still suffer from generating videos with incorrect and jittery motion, which severely restricts their applications in practice.

To address the aforementioned challenges, we propose CONSISTI2V, a simple yet effective framework capable of enhancing the visual consistency for I2V generation. Our method focuses on improving the first frame conditioning mechanisms in the I2V model and optimizing the inference noise initialization during sampling. To produce videos that closely resemble the first frame, we apply cross-frame attention mechanisms in the model's spatial layers to achieve fine-grained *spatial* first frame conditioning. To ensure the temporal smoothness and coherency of the generated video, we include a local window of the first frame features in the *temporal* layers to augment their attention operations. During inference, we propose FrameInit, which leverages the low-frequency component of the first frame image and combines it with the initial noise to act as a *layout* guidance and eliminate the noise discrepancy between training and inference. By integrating these design optimizations, our model generates highly consistent videos and can be easily extended to other applications such as autoregressive long video generation and camera motion control. Our model achieves state-of-the-art results on public I2V generation benchmarks. We further conduct extensive automatic and human evaluations on a self-collected dataset I2V-Bench to verify the effectiveness of our method for I2V generation. Our contributions are summarized below:

1. We introduce CONSISTI2V, a diffusion-based model that performs spatiotemporal conditioning over the first frame to enhance the visual consistency in video generation.
2. We devise FrameInit, an inference-time noise initialization strategy that uses the low-frequency band from the first frame to stabilize video generation. FrameInit can also support applications such as autoregressive long video generation and camera motion control.
3. We propose I2V-Bench, a comprehensive quantitative evaluation benchmark dedicated to evaluating I2V generation models. We will release our evaluation dataset to foster future I2V generation research.

---

[2]In this study, we follow prior work (Zhang et al., 2023a) and focus on text-guided I2V generation.

## 2 Related Work

**Text-to-Video Generation** Recent studies in T2V generation has evolved from using GAN-based models (Fox et al., 2021; Brooks et al., 2022; Tian et al., 2021) and autoregressive transformers (Ge et al., 2022; Hong et al., 2022) to embracing diffusion models. Current methods usually extend T2I generation frameworks to model video data. VDM (Ho et al., 2022b) proposes a space-time factorized U-Net (Ronneberger et al., 2015) and interleaved temporal attention layers to enable video modelling. Imagen-Video (Ho et al., 2022a) and Make-A-Video (Singer et al., 2022) employ pixel-space diffusion models Imagen (Saharia et al., 2022) and DALL-E 2 (Ramesh et al., 2022) for high-definition video generation. Another line of work (He et al., 2022; Chen et al., 2023a; Khachatryan et al., 2023; Guo et al., 2023) generate videos with latent diffusion models (Rombach et al., 2022) due to the high efficiency of LDMs. In particular, MagicVideo (Zhou et al., 2022) inserts simple adaptor layers and Latent-Shift (An et al., 2023) utilizes temporal shift modules (Lin et al., 2019) to enable temporal modelling. Subsequent works generally follow VideoLDM (Blattmann et al., 2023) and insert temporal convolution and attention layers inside the LDM U-Net for video generation.

Another research stream focuses on optimizing the noise initialization for video generation. PYoCo (Ge et al., 2023) proposes a video-specific noise prior and enables each frame's initial noise to be correlated with other frames. FreeNoise (Qiu et al., 2023) devises a training-free noise rescheduling method for long video generation. FreeInit (Wu et al., 2023c) leverages the low-frequency component of a noisy video to eliminate the initialization gap between training and inference of the diffusion models.

**Video Editing** As paired video data before and after editing is hard to obtain, several methods (Ceylan et al., 2023; Wu et al., 2023b; Geyer et al., 2023; Wu et al., 2023a; Zhang et al., 2023b; Cong et al., 2023) employ a pretrained T2I model for zero-shot/few-shot video editing. To ensure temporal coherency between individual edited frames, these methods apply cross-frame attention mechanisms in the T2I model. Specifically, Tune-A-Video (Wu et al., 2023b) and Pix2Video (Ceylan et al., 2023) modify the T2I model's self-attention layers to enable each frame to attend to its immediate previous frame and the video's first frame. TokenFlow (Geyer et al., 2023) and Fairy (Wu et al., 2023a) select a set of anchor frames such that all anchor frames can attend to each other during self-attention.

**Image-to-Video Generation** Steering the video's content using only text descriptions can be challenging. Recently, a myriad of methods utilizing both first frame and text for I2V generation have emerged as a solution to achieve more controllable video generation. Among these methods, Emu-Video (Girdhar et al., 2023), SEINE (Chen et al., 2023c), AnimateAnything (Dai et al., 2023) and PixelDance (Zeng et al., 2023) propose simple modifications to the T2V U-Net by concatenating the latent features of the first frame with input noise to enable first-frame conditioning. I2VGen-XL (Zhang et al., 2023a), Dynamicrafter (Xing et al., 2023) and Moonshot (Zhang et al., 2024) add extra image cross-attention layers in the model to inject stronger conditional signals into the generation process. Our approach varies from previous studies in two critical respects: (1) our spatiotemporal feature conditioning methods effectively leverage the input first frame, resulting in better visual consistency in the generated videos and enabling efficient training on public video-text datasets. (2) We develop noise initialization strategies in the I2V inference processes, while previous I2V generation works rarely focus on this aspect.

## 3 Methodology

Given an image $x^1 \in \mathbb{R}^{C \times H \times W}$ and a text prompt $\mathbf{s}$, the goal of our model is to generate an $N$ frame video clip $\hat{\mathbf{x}} = \{x^1, \hat{x}^2, \hat{x}^3, ... \hat{x}^N\} \in \mathbb{R}^{N \times C \times H \times W}$ such that $x^1$ is the first frame of the video, and enforce the appearance of the rest of the video to be closely aligned with $x^1$ and the content of the video to follow the textual description in $\mathbf{s}$. We approach this task by employing first-frame conditioning mechanisms in spatial and temporal layers of our model and applying layout-guided noise initialization during inference. The overall model architecture and inference pipeline of ConsistI2V are shown in Figure 2.

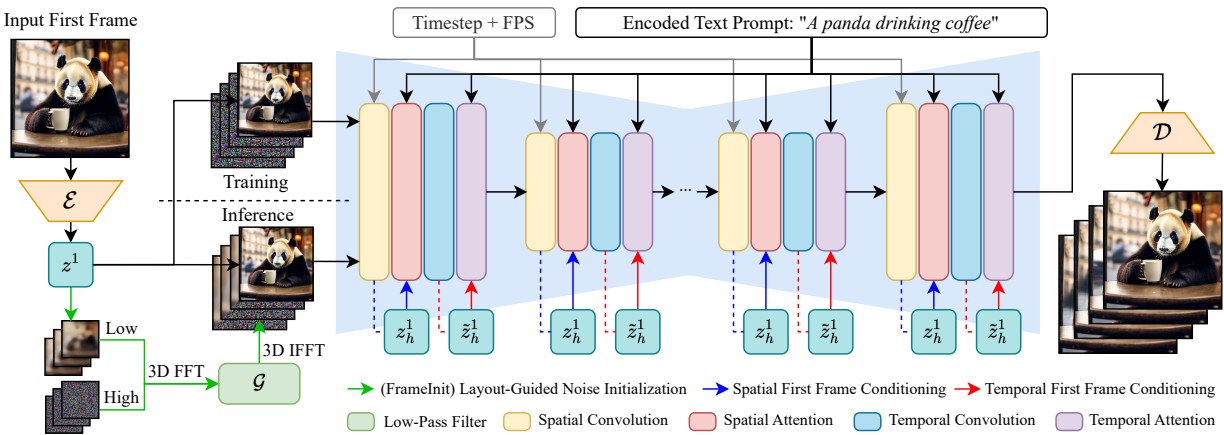

Figure 2: Our CONSISTI2V framework. In our model, we concatenate the first frame latent $z^1$ to the input noise and perform first frame conditioning by augmenting the spatial and temporal self-attention operations in the model with the intermediate hidden states $z_h^1$. During inference, we incorporate the low-frequency component from $z^1$ to initialize the inference noise and guide the video generation process.

## 3.1 Preliminaries

**Diffusion Models (DMs)** (Sohl-Dickstein et al., 2015; Ho et al., 2020) are generative models that learn to model the data distribution by iteratively recovering perturbed inputs. Given a training sample $\mathbf{x_0} \sim q(\mathbf{x_0})$, DMs first obtain the corrupted input through a forward diffusion process $q(\mathbf{x}_t|\mathbf{x}_0, t), t \in \{1, 2, ..., T\}$ by using the parameterization trick from Sohl-Dickstein et al. (2015) and gradually adds Gaussian noise $\epsilon \in \mathcal{N}(\mathbf{0}, \mathbf{I})$ to the input: $\mathbf{x}_t = \sqrt{\bar{\alpha}_t}\mathbf{x}_0 + \sqrt{1 - \bar{\alpha}_t}\epsilon, \bar{\alpha}_t = \prod_{i=1}^{t}(1 - \beta_i)$, where $0 < \beta_1 < \beta_2 < ... < \beta_T < 1$ is a known variance schedule that controls the amount of noise added at each time step $t$. The diffusion model is then trained to approximate the backward process $p(\mathbf{x}_{t-1}|\mathbf{x}_t)$ and recovers $\mathbf{x}_{t-1}$ from $\mathbf{x}_t$ using a denoising network $\epsilon_\theta(\mathbf{x}_t, \mathbf{c}, t)$, which can be learned by minimizing the mean squared error (MSE) between the predicted and target noise: $\min_\theta \mathbb{E}_{\mathbf{x}, \epsilon \in \mathcal{N}(\mathbf{0}, \mathbf{I}), \mathbf{c}, t}[\|\epsilon - \epsilon_\theta(\mathbf{x}_t, \mathbf{c}, t)\|_2^2]$ ($\epsilon-$prediction). Here, $\mathbf{c}$ denotes the (optional) conditional signal that DMs can be conditioned on. For our model, $\mathbf{c}$ is a combination of an input first frame and a text prompt.

**Latent Diffusion Models (LDMs)** (Rombach et al., 2022) are variants of diffusion models that first use a pretrained encoder $\mathcal{E}$ to obtain a latent representation $\mathbf{z}_0 = \mathcal{E}(\mathbf{x}_0)$. LDMs then perform the forward process $q(\mathbf{z}_t|\mathbf{z}_0, t)$ and the backward process $p_\theta(\mathbf{z}_{t-1}|\mathbf{z}_t)$ in this compressed latent space. The generated sample $\hat{\mathbf{x}}$ can be obtained from the denoised latent using a pretrained decoder $\hat{\mathbf{x}} = \mathcal{D}(\hat{\mathbf{z}})$.

## 3.2 Model Architecture

**U-Net Inflation for Video Generation**  Our model is developed based on text-to-image (T2I) LDMs (Rombach et al., 2022) that employ the U-Net (Ronneberger et al., 2015) model for image generation. This U-Net model contains a series of spatial downsampling and upsampling blocks with skip connections. Each down/upsampling block is constructed with two types of basic blocks: spatial convolution and spatial attention layers. We insert a 1D temporal convolution block after every spatial convolution block and temporal attention blocks at certain attention resolutions to make it compatible with video generation tasks. Our temporal convolution and attention blocks share the exact same architecture as their spatial counterparts, apart from the convolution and attention operations are operated along the temporal dimension. We incorporate RoPE (Su et al., 2024) embeddings to represent positional information in the temporal layers and employ the PYoCo (Ge et al., 2023) *mixed* noise prior for noise initialization (*cf.* Appendix A.1).

**First Frame Condition Injection**  We leverage the variational autoencoder (VAE) (Kingma & Welling, 2013) of the T2I LDM to encode the input first frame into latent representation $z^1 = \mathcal{E}(x^1) \in \mathbb{R}^{C' \times H' \times W'}$

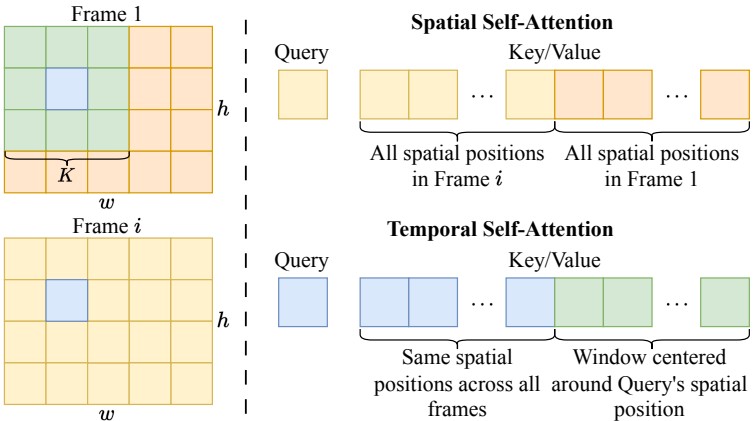

Figure 3: Visualization of our proposed spatial and temporal first frame conditioning schemes. For spatial self-attention layers, we employ cross-frame attention mechanisms and expand the keys and values with the features from all spatial positions in the first frame. For temporal self-attention layers, we augment the key and value vectors with a local feature window from the first frame.

and use $z^1$ as the conditional signal. To inject this signal into our model, we directly replace the first frame noise $\epsilon^1$ with $z^1$ and construct the model input as $\hat{\epsilon} = \{z^1, \epsilon^2, \epsilon^3, ..., \epsilon^N\} \in \mathbb{R}^{N \times C' \times H' \times W'}$.

### 3.3 Fine-Grained Spatial Feature Conditioning

The spatial attention layer in the LDM U-Net contains a self-attention layer that operates on each frame independently and a cross-attention layer that operates between frames and the encoded text prompt. Given an intermediate hidden state $z^i$ of the $i^{\text{th}}$ frame, the self-attention operation is formulated as the attention between different spatial positions of $z^i$:

$$Q_s = W_s^Q z^i, K_s = W_s^K z^i, V_s = W_s^V z^i, \tag{1}$$

$$\text{Attention}(Q_s, K_s, V_s) = \text{Softmax}(\frac{Q_s K_s^\top}{\sqrt{d}})V_s, \tag{2}$$

where $W_s^Q, W_s^K$ and $W_s^V$ are learnable projection matrices for creating query, key and value vectors from the input. $d$ is the dimension of the query and key vectors. To achieve better visual coherency in the video, we modify the key and value vectors in the self-attention layers to also include features from the first frame $z^1$ (*cf.* Figure 3):

$$Q_s = W_s^Q z^i, K_s' = W_s^K[z^i, z^1], V_s' = W_s^V[z^i, z^1], \tag{3}$$

where $[\cdot]$ represents the concatenation operation such that the token sequence length in $K_s'$ and $V_s'$ are doubled compared to the original $K_s$ and $V_s$. In this way, each spatial position in all frames gets access to the complete information from the first frame, allowing fine-grained feature conditioning in the spatial attention layers.

### 3.4 Window-based Temporal Feature Conditioning

The temporal self-attention operations in video diffusion models (Ho et al., 2022b; Blattmann et al., 2023; Wang et al., 2023d; Chen et al., 2023a) share a similar formulation to spatial self-attention (*cf.* Equation 1), with the exception that the intermediate hidden state being formulated as $\bar{\mathbf{z}} \in \mathbb{R}^{(H \times W) \times N \times C}$, where $N$ is the number of frames and $C, H, W$ correspond to the channel, height and width dimension of the hidden state tensor. Formally, given an input hidden state $\mathbf{z} \in \mathbb{R}^{N \times C \times H \times W}$, $\bar{\mathbf{z}}$ is obtained by reshaping the height and width dimension of $\mathbf{z}$ to the batch dimension such that $\bar{\mathbf{z}} \leftarrow \texttt{rearrange}(\mathbf{z}, \texttt{"N C H W -> (H W) N C"})$ (using `einops` (Rogozhnikov, 2021) notation). Intuitively, every $1 \times N \times C$ matrices in $\bar{\mathbf{z}}$ represents features at the same $1 \times 1$ spatial location across all $N$ frames. This limits the temporal attention layers to always look at a small spatial window when attending to features between different frames.

To effectively leverage the first frame features, we also augment the temporal self-attention layers to include a broader window of features from the first frame. We propose to compute the query, key and value from $\bar{\mathbf{z}}$ as:

$$Q_t = W_t^Q \bar{\mathbf{z}}, K_t' = W_t^K[\bar{\mathbf{z}}, \tilde{z}^1], V_t' = W_t^V[\bar{\mathbf{z}}, \tilde{z}^1], \tag{4}$$

where $\tilde{z}^1 \in \mathbb{R}^{(H \times W) \times (K \times K - 1) \times C}$ is a tensor constructed in a way that its $h \times w$ position in the batch dimension corresponds to an $K \times K$ window of the first frame features, centred at the spatial position of $(h, w)$. The first frame feature vector at $(h, w)$ is not included in $\tilde{z}^1$ as it is already presented in $\bar{\mathbf{z}}$, making the second dimension of $\tilde{z}^1$ as $K \times K - 1$. We pad $\mathbf{z}$ in its spatial dimensions by replicating the boundary values to ensure that all spatial positions in $\mathbf{z}$ will have a complete window around them. We then concatenate $\bar{\mathbf{z}}$ with $\tilde{z}^1$ to enlarge the sequence length of the key and value matrices. See Figure 3 for a visualization of the augmented temporal self-attention. Our main rationale is that the visual objects in a video may move to different spatial locations as the video progresses. The vanilla or common formulation of temporal self-attention in video diffusion models essentially assumes $K = 1$, as shown by the blue tokens in Figure 3. Having an extra window of keys and values around the query location with the window size $K > 1$ (green tokens in Figure 3) increases the probability of attending to the same entity in the first frame when performing temporal self-attention. In practice, we set $K = 3$ to control the time complexity of the attention operations. As the input latent is downsampled $8\times$ by the VAE, and the denoising U-Net further downsamples the feature map in its intermediate layers, our selected window size ($K = 3$) can still create a large receptive field in deeper U-Net layers. For example, in the second last level of the U-Net ($32\times$ total downsampling), a $256 \times 256$ video will be compressed into a $8 \times 8$ feature map. Consequently, a $3 \times 3$ window on this feature map covers a region of $96 \times 96$ in the input first frame. This is a significant increase from previous methods where a $1 \times 1$ window on the feature map corresponded to only a $24 \times 24$ region in the original frame.

### 3.5 Inference-time Layout-Guided Noise Initialization

Existing literature (Lin et al., 2024) in image diffusion models has identified that there exists a noise initialization gap between training and inference, due to the fact that common diffusion noise schedules create an information leak to the diffusion noise during training, causing it to be inconsistent with the random Gaussian noise sampled during inference. In the domain of video generation, this initialization gap has been further explored by FreeInit (Wu et al., 2023c), showing that the information leak mainly comes from the low-frequency component of a video after spatiotemporal frequency decomposition and adding this low-frequency component to the initial inference noise greatly enhances the quality of the generated videos. Inspired by the significance of frequency bands in images and videos for human perception, prior work like Blurring Diffusion Models (Hoogeboom & Salimans, 2022) employs diffusion processes in the frequency domain to enhance generation results. Here, we also aim to study how to effectively leverage the video's different frequency components to enhance I2V generation.

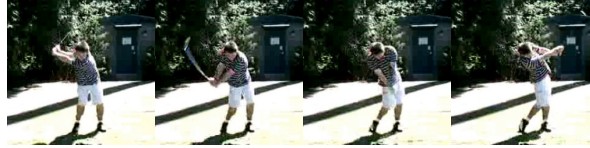

Reconstructed sample from the original latent $\mathbf{z}_0$

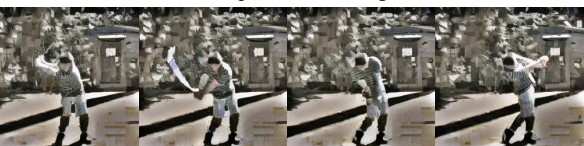

Reconstructed sample from the high-frequency component of $\mathbf{z}_0$

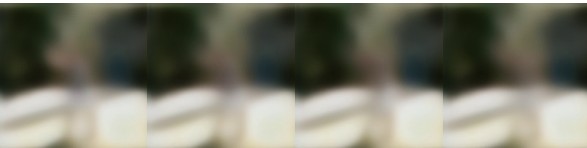

Reconstructed sample from the low-frequency component of $\mathbf{z}_0$

Figure 4: Different frequency bands from the original latent $\mathbf{z}_0$ after spatiotemporal frequency decomposition.

To better understand how spatiotemporal frequency bands and visual features are correlated in videos, we visualize the videos decoded from the VAE latents after spatiotemporal frequency decomposition, as shown in Figure 4. We observe that the video's high-frequency component captures the fast-moving objects and the fine details in the video, whereas the low-frequency component corresponds to those slowly moving parts and represents an overall layout. Based on this observation, we propose FrameInit, which duplicates the input first frame into a static video and uses its low-frequency component as a coarse layout guidance during inference.

Formally, given the latent representation $\mathbf{z}_0$ of a static video and an inference noise $\epsilon$, we first add $\tau$ step inference noise to the static video to obtain $\mathbf{z}_\tau = \texttt{add\_noise}(\mathbf{z}_0, \epsilon, \tau)$. We then extract the low-frequency component of $\mathbf{z}_\tau$ and mix it with $\epsilon$:

$$\mathcal{F}^{low}_{\mathbf{z}_\tau} = \texttt{FFT\_3D}(\mathbf{z}_\tau) \odot \mathcal{G}(D_0), \tag{5}$$

$$\mathcal{F}^{high}_{\epsilon} = \texttt{FFT\_3D}(\epsilon) \odot (1 - \mathcal{G}(D_0)), \tag{6}$$

$$\epsilon' = \texttt{IFFT\_3D}(\mathcal{F}^{low}_{\mathbf{z}_\tau} + \mathcal{F}^{high}_{\epsilon}), \tag{7}$$

where $\texttt{FFT\_3D}$ is the 3D discrete fast Fourier transformation operating on spatiotemporal dimensions and $\texttt{IFFT\_3D}$ is the inverse FFT operation. $\mathcal{G}$ is the Gaussian low-pass filter parameterized by the normalized space-time stop frequency $D_0$. The modified noise $\epsilon'$ containing the low-frequency information from the static video is then used for denoising.

By implementing FrameInit, our empirical analysis reveals a significant enhancement in the stabilization of generated videos, demonstrating improved video quality and consistency. FrameInit also enables our model with two additional applications: (1) autoregressive long video generation and (2) camera motion control. We showcase more results for each application in Section 5.5.

## 4   I2V-Bench

Existing video generation benchmarks such as UCF-101 (Soomro et al., 2012) and MSR-VTT (Xu et al., 2016) fall short in video resolution, diversity, and aesthetic appeal. To bridge this gap, we introduce the I2V-Bench evaluation dataset, featuring 2,950 high-quality YouTube videos curated based on strict resolution and aesthetic standards. We organized these videos into 16 distinct categories, such as Scenery, Sports, Animals, and Portraits. Further details are available in the Appendix.

**Evaluation Metrics**   Following VBench (Huang et al., 2023b), our evaluation framework encompasses two key dimensions, each addressing distinct aspects of I2V performances: (1) **Visual Quality** assesses the perceptual quality of the video output regardless of the input prompts. We measure the subject and background consistency, temporal flickering, motion smoothness and dynamic degree. (2) **Visual Consistency** evaluates the video's adherence to the text prompt given by the user. We measure object consistency, scene consistency and overall video-text consistency. Further details can be found in Appendix C.

## 5   Experiments

### 5.1   Implementation Details

We use Stable Diffusion 2.1-base (Rombach et al., 2022) as the base T2I model to initialize ConsistI2V and train the model on the WebVid-10M (Bain et al., 2021) dataset, which contains $\sim$10M video-text pairs. For each video, we sample 16 frames with a spatial resolution of $256 \times 256$ and a frame interval between $1 \leq v \leq 5$, which is used as a conditional input to the model to enable FPS control. We use the first frame as the image input and learn to denoise the subsequent 15 frames during training. Our model is trained with the $\epsilon$ objective over all U-Net parameters using a batch size of 192 and a learning rate of 5e-5 for 170k steps. During training, we randomly drop input text prompts with a probability of 0.1 to enable classifier-free guidance (Ho & Salimans, 2022). During inference, we employ the DDIM sampler (Song et al., 2020) with 50 steps and classifier-free guidance with a guidance scale of $w = 7.5$ to sample videos. We apply FrameInit with $\tau = 850$ and $D_0 = 0.25$ for inference noise initialization.

### 5.2   Quantitative Evaluation

**UCF-101 & MSR-VTT**   We evaluate ConsistI2V on two public datasets UCF-101 (Soomro et al., 2012) and MSR-VTT (Xu et al., 2016). We report Fréchet Video Distance (FVD) (Unterthiner et al., 2019) and Inception Score (IS) (Salimans et al., 2016) for video quality assessment, Fréchet Inception Distance (FID) (Heusel et al., 2017) for frame quality assessment and CLIP similarity (CLIPSIM) (Wu et al., 2021) for

Table 1: Quantitative evaluation results for ConsistI2V. †: the statistics also include the data for training the base video generation model. **Bold:** best results. Underline: second best.

| Method | #Data | UCF-101 | | | MSR-VTT | | Human Eval: Consistency | |
| | | FVD ↓ | IS ↑ | FID ↓ | FVD ↓ | CLIPSIM ↑ | Appearance ↑ | Motion ↑ |
|---|---|---|---|---|---|---|---|---|
| AnimateAnything | 10M+20K† | 642.64 | **63.87** | **10.00** | 218.10 | 0.2661 | 43.07% | 20.26% |
| I2VGen-XL | 35M | 597.42 | 18.20 | 42.39 | 270.78 | 0.2541 | 1.79% | 9.43% |
| DynamiCrafter | 10M+10M† | 404.50 | 41.97 | 32.35 | 219.31 | 0.2659 | 44.49% | 31.10% |
| SEINE | 25M+10M† | 306.49 | 54.02 | 26.00 | 152.63 | **0.2774** | 48.16% | 36.76% |
| ConsistI2V | 10M | **177.66** | 56.22 | 15.74 | **104.58** | 0.2674 | **53.62%** | **37.04%** |

video-text alignment evaluation. We refer readers to Appendix B.2 for the implementation details of these metrics. We evaluate FVD, FID and IS on UCF-101 over 2048 videos and FVD and CLIPSIM on MSR-VTT's test split (2990 samples). We focus on evaluating the I2V generation capability of our model: given a video clip from the evaluation dataset, we randomly sample a frame and use it along with the text prompt as the input to our model. All evaluations are performed in a zero-shot manner.

We compare ConsistI2V against four open-sourced I2V generation models: I2VGen-XL (Zhang et al., 2023a), AnimateAnything (Dai et al., 2023), DynamiCrafter (Xing et al., 2023) and SEINE (Chen et al., 2023c). Quantitative evaluation results are shown in Table 1. We observe that while AnimateAnything achieves better IS and FID, its generated videos are mostly near static (see Figure 5 for visualizations), which severely limits the video quality (highest FVD of 642.64). Our model significantly outperforms the rest of the baseline models in all metrics, except for CLIPSIM on MSR-VTT, where the result is slightly lower than SEINE. We note that SEINE, initialized from LaVie (Wang et al., 2023d), benefited from a larger and superior-quality training dataset, including Vimeo25M, WebVid-10M, and additional private datasets. In contrast, ConsistI2V is directly initialized from T2I models and only trained on WebVid-10M, showcasing our method's effectiveness.

Table 2: Automatic evaluation results on I2V-Bench. Consist. denotes the consistency metrics. **Bold:** best results. Underline: second best.

| Method | Temporal Flickering ↑ | Motion Smoothness ↑ | Dynamic Degree ↑ | Background Consist. ↑ | Subject Consist. ↑ | Object Consist. ↑ | Scene Consist. ↑ | Overall Consist. ↑ |
|---|---|---|---|---|---|---|---|---|
| AnimateAnything | **99.08** | **99.23** | 3.69 | **98.50** | **97.90** | 32.56 | 27.18 | 18.74 |
| I2VGen-XL | 94.03 | 96.03 | 57.88 | 94.52 | 89.36 | 29.25 | 23.50 | 16.89 |
| DynamiCrafter | 93.81 | 95.89 | **64.11** | 93.47 | 88.01 | 32.18 | 23.93 | 18.68 |
| SEINE | 96.08 | 97.92 | 42.12 | 93.98 | 88.21 | **35.43** | **27.68** | **20.21** |
| ConsistI2V | 96.65 | 97.77 | 37.48 | 94.69 | 90.85 | 32.06 | 24.81 | 19.50 |

**I2V-Bench**  We present the automatic evaluation results for ConsistI2V and the baseline models in Table 2. Similar to previous findings, we observe that AnimateAnything achieves the best motion smoothness and appearance consistency among all models. However, it significantly falls short in generating videos with higher motion magnitude, registering a modest dynamic degree value of only 3.69 (visual results as shown in Figure 5). On the other hand, our model achieves a better balance between motion magnitude and video quality, outperforms all other baseline models excluding AnimateAnything in terms of motion quality (less flickering and better smoothness) and visual consistency (higher background/subject consistency) and achieves a competitive overall video-text consistency.

**Human Evaluation**  To further validate the generation quality of our model, we conduct a human evaluation based on 548 samples from our ConsistI2V and the baseline models. We randomly distribute a subset of samples to each participant, presenting them with the input image, text prompt, and all generated videos. Participants are then asked to answer two questions: to identify the videos with the best overall appearance and motion consistency. Each question allows for one or more selections. We collect a total of 1061 responses from 13 participants and show the results in the right part of Table 1. As demonstrated by the results, our

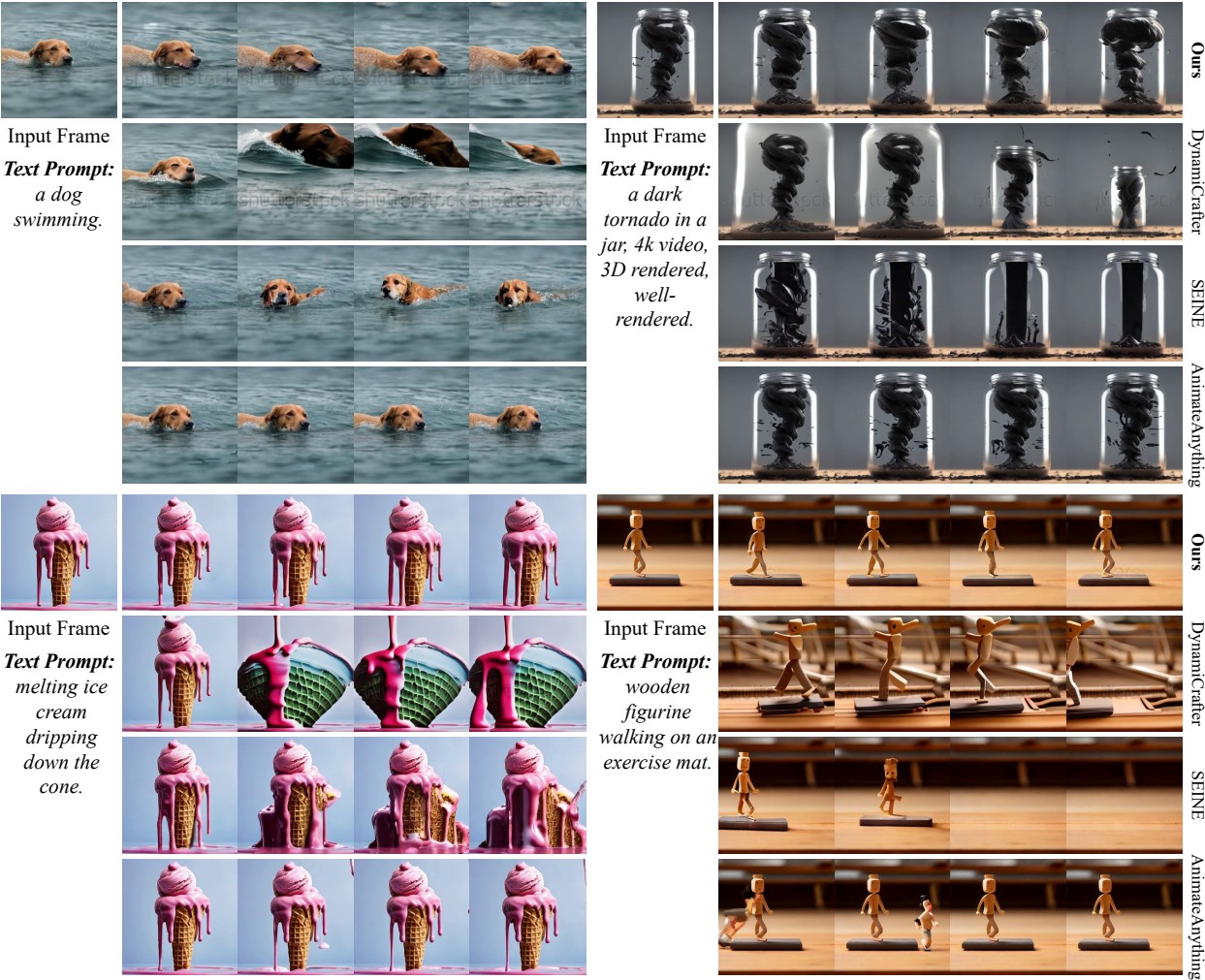

Figure 5: Qualitative comparisons between DynamiCrafter, SEINE, AnimateAnything and our ConsistI2V. Input first frames are generated by PixArt-$\alpha$ (Chen et al., 2023b) and SDXL (Podell et al., 2023).

model ranked top in both metrics, achieving a comparable motion consistency with SEINE and a significantly higher appearance consistency than all other baseline models.

### 5.3 Qualitative Evaluation

We present a visual comparison of our model with DynamiCrafter, SEINE and AnimateAnything in Figure 5. We exclude I2VGen-XL in this section as its generated video cannot fully adhere to the visual details from the input first frame. As shown in the figure, current methods often struggle with maintaining appearance and motion consistency in video sequences. This can include (1) sudden changes in subject appearance mid-video, as demonstrated in the "ice cream" case for DynamiCrafter and SEINE; (2) background inconsistency, as observed in DynamiCrafter's "wooden figurine walking" case; (3) unnatural object movements, evident in the "dog swimming" case (DynamiCrafter) and "tornado in a jar" case (SEINE) and (4) minimal or absent movement, as displayed in most of the generated videos by AnimateAnything. On the other hand, ConsistI2V produces videos with subjects that consistently align with the input first frame. Additionally, our generated videos exhibit more natural and logical motion, avoiding abrupt changes and thereby ensuring improved appearance and motion consistency. More visual results of our model can be found in Appendix H.

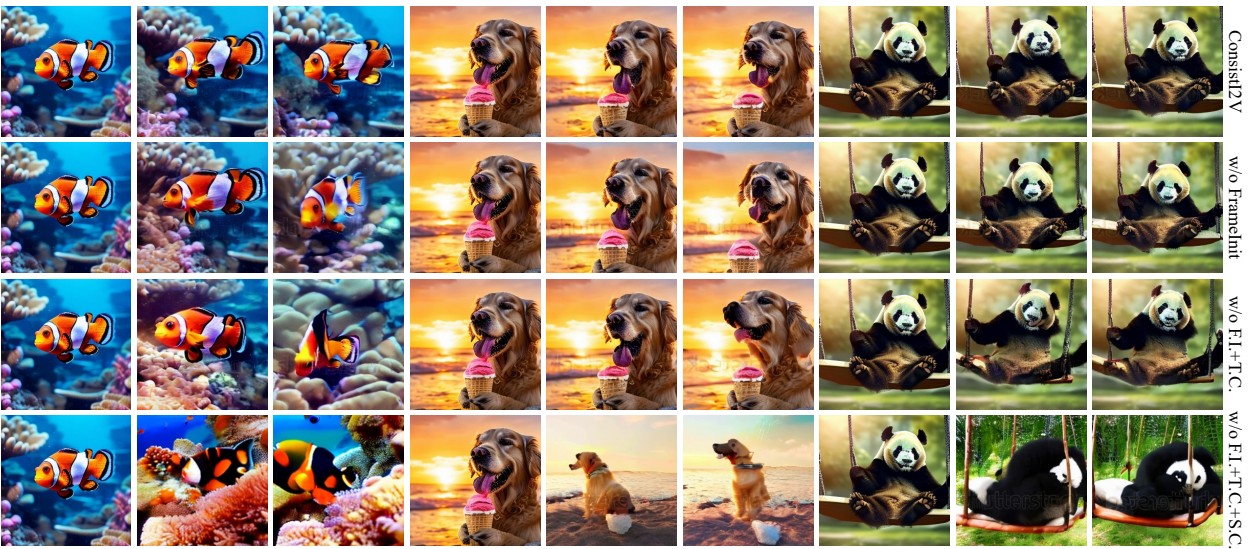

**Text Prompt:** *clown fish swimming through the coral reef.*

**Text Prompt:** *a golden retriever eating ice cream on a beautiful tropical beach at sunset.*

**Text Prompt:** *a panda playing on a swing set.*

Figure 6: Visual comparisons of our method after disabling FrameInit (F.I.), temporal conditioning (T.C.) and spatial conditioning (S.C.). We use the same seed to generate all videos.

Table 3: UCF-101 ablation study results and runtime statistics for spatiotemporal first frame conditioning and FrameInit. T.Cond. and S.Cond. correspond to temporal and spatial first frame conditioning, respectively.

|  | UCF-101 (Zero-Shot) | | | Inference Cost Statistics | | |
|---|---|---|---|---|---|---|
|  | FVD ↓ | IS ↑ | FID ↓ | GPU Mem. | Runtime | TFLOPs (UNet) |
| ConsistI2V | 177.66 | 56.22 | 15.74 | 9249 Mb | 18.48s | 5.168 |
| w/o FrameInit | 245.79 | 42.21 | 24.08 | 9245 Mb | 18.37s | 5.168 |
| w/o FrameInit & T.Cond. | 224.16 | 42.98 | 24.04 | 9233 Mb | 17.93s | 5.117 |
| w/o FrameInit & T.Cond. & S.Cond. | 704.48 | 21.97 | 68.39 | 9017 Mb | 17.14s | 5.015 |

## 5.4 Ablation Studies

To verify the effectiveness of our design choices, we conduct an ablation study on UCF-101 by iteratively disabling FrameInit, temporal first frame conditioning and spatial first frame conditioning. We follow the same experiment setups in Section 5.2 and show the results in Table 3.

**Effectiveness of FrameInit** According to the results, we find that applying FrameInit greatly enhances the model performance for all metrics. Our empirical observation suggests that FrameInit can stabilize the output video and reduce the abrupt appearance and motion changes. As shown in Figure 6, while our model can still generate reasonable results without enabling FrameInit, the output videos also suffer from a higher chance of rendering sudden object movements and blurry frames (last frame in the second row). This highlights the effectiveness of FrameInit in producing more natural motion and higher-quality frames.

**Hyperparameters of FrameInit** To further investigate how the FrameInit hyperparameters $\tau$ (number of steps of inference noise added to the first frame) and $D_0$ (spatiotemporal stop frequency) affect the generated videos, we conduct an additional qualitative experiment by generating the same video using different FrameInit hyperparameters and show the results in Figure 7. Comparing (1) and (4) in Figure 7, we find that while using our default setting of FrameInit achieves more stable outputs, it also slightly restricts the motion magnitude of the generated video compared to not using FrameInit at all. From comparisons between (1), (5), and (1), (6), we observe that either increasing $D_0$ alone or reducing $\tau$ alone does not significantly impact the motion magnitude of the videos. However, comparisons between (1), (2), and (3) reveal that when both $D_0$ is increased and $\tau$ is decreased, there is a further restriction in video dynamics. This suggests that the static first frame video exerts a stronger layout guidance, intensifying its influence on video dynamics.

(1) $\tau = 850$, $D_0 = 0.25$ *(Default)*                       (4) w/o FrameInit

(2) $\tau = 750$, $D_0 = 0.50$                              (5) $\tau = 850$, $D_0 = 0.75$

(3) $\tau = 650$, $D_0 = 0.75$                              (6) $\tau = 650$, $D_0 = 0.25$

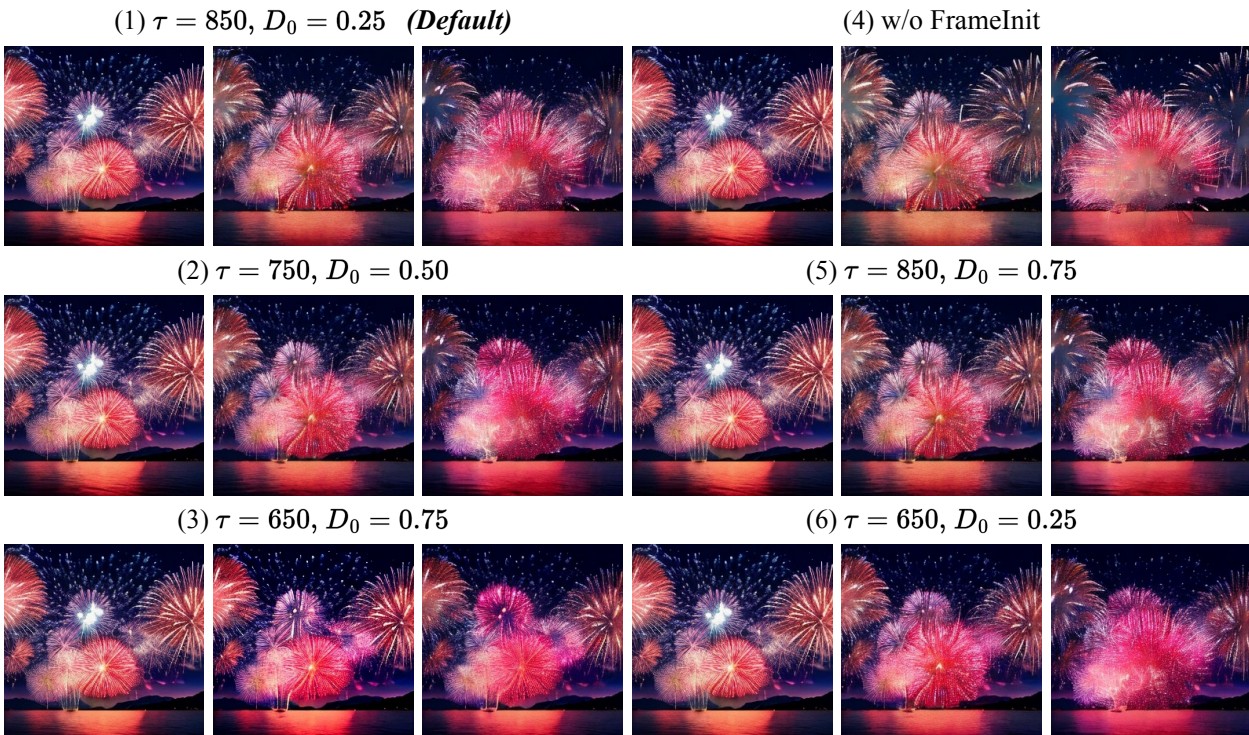

**Text Prompt:** *fireworks bloom in the night sky.*

Figure 7: Visual comparisons of employing different FrameInit hyperparameters during I2V generation. All videos are generated using the same seed.

**Spatiotemporal First Frame Conditioning** Our ablation results on UCF-101 (*c.f.* Table 3) reflects a significant performance boost after applying the proposed spatial and temporal first frame conditioning mechanisms to our model. Although removing temporal first frame conditioning leads to an overall better performance for the three quantitative metrics, in practice we find that only using spatial conditioning often results in jittering motion and larger object distortions, evident in the third row of Figure 6. When both spatial and temporal first frame conditioning is removed, our model loses the capability of maintaining the appearance of the input first frame.

**Runtime Efficiency** To better understand how the proposed model components affect the inference cost, we also benchmark the runtime statistics of our ConsistI2V and its variants after disabling different components. We measure all models on a single Nvidia RTX 4090 with float32 and an inference batch size of 1. For TFLOPs computation, we only consider the denoising U-Net and exclude components such as the VAE and text encoder as these modules share the same computation across all model variants. As shown in the right part of Table 3, we find that the proposed spatiotemporal feature conditioning and FrameInit incur limited computational overhead during inference, introducing approximately 200Mb extra GPU memory and ∼1.3s additional inference time. These results demonstrate that our method is highly efficient during inference.

### 5.5 More Applications

**Autoregressive Long Video Generation** While our I2V model provides native support for long video generation by reusing the last frame of the previous video to generate the next video, we observe that directly using the model to generate long videos may lead to suboptimal results, as the artifacts in the previous video clip will often accumulate throughout the autoregressive generation process. We find that using FrameInit to guide the generation of each video chunk helps stabilize the autoregressive video generation process and results in a more consistent visual appearance throughout the video, as shown in Figure 8.

**Camera Motion Control** When adopting FrameInit for inference, instead of using the static first frame video as the input, we can alternatively create synthetic camera motions from the first frame and use it as the layout condition. For instance, camera panning can be simulated by creating spatial crops in the first frame starting from one side and gradually moving to the other side. As shown in the first two examples under camera motion control in Figure 8, by simply tweaking the FrameInit parameters to $\tau = 750$ and $D_0 = 0.5$ and using the synthetic camera motion as the layout guidance, we are able to achieve camera panning and zoom-in/zoom-out effects in the generated videos without any additional training. For camera motions that involve perspective view change (e.g. rotating), we propose to apply image warping to the input first frame to create a sequence of perspective transformations. We then use this synthetic perspective view change as the layout condition to generate new videos. As shown in the last example of Figure 8, we can synthesize camera rotation by creating synthetic perspective view changes based on the coordinates of four selected key points in the first frame. As perspective view changes are often more complex than 2D panning/zooming, the FrameInit hyperparameters for this example are selected as $\tau = 700$ and $D_0 = 0.5$ to enforce a stronger layout guidance.

# 6 Limitations

Our current method has several limitations: (1) our training dataset WebVid-10M (Bain et al., 2021) predominantly comprises low-resolution videos, and a consistent feature across this dataset is the presence of a watermark, located at a fixed position in all the videos. As a result, our generated videos will also have a high chance of getting corrupted by the watermark, and we currently only support generating videos at a relatively low resolution. (2) While our proposed FrameInit enhances the stability of the generated videos, we also observe that our model sometimes creates videos with limited motion magnitude, thereby restricting the subject movements in the video content. (3) Our spatial first frame conditioning method requires tuning the spatial U-Net layers during training, which limits the ability of our model to directly adapt to personalized T2I generation models and increases the training costs. (4) Our model shares some other common limitations with the base T2I generation model Stable Diffusion (Rombach et al., 2022), such as not being able to correctly render human faces and legible text.

# 7 Conclusion

We presented CONSISTI2V, an I2V generation framework designed to improve the visual consistency of

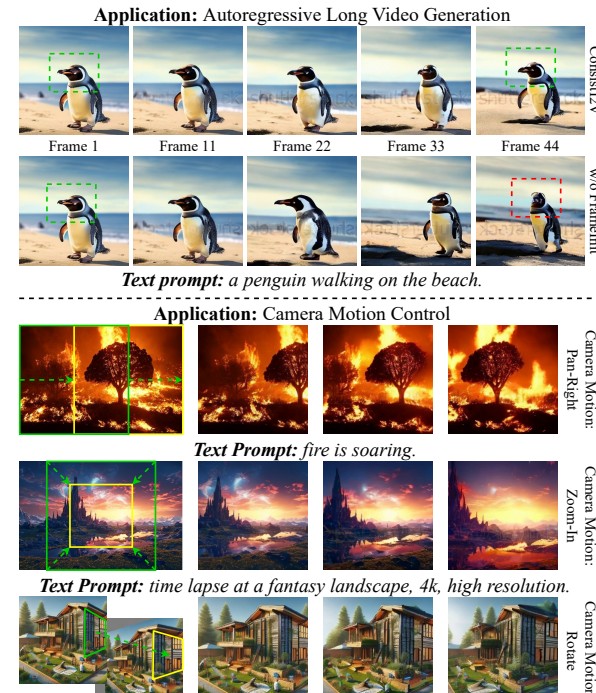

Figure 8: Applications of CONSISTI2V. **Upper panel:** FrameInit enhances object consistency in long video generation. **Lower panel:** FrameInit enables training-free camera motion control.

generated videos by integrating our novel spatiotemporal first frame conditioning and FrameInit layout guidance mechanisms. Our approach enables the generation of highly consistent videos and supports applications including autoregressive long video generation and camera motion control. We conducted extensive automatic and human evaluations on various benchmarks, including our proposed I2V-Bench and demonstrated exceptional I2V generation results. For future work, we plan to refine our training paradigm and incorporate higher-quality training data to further scale up the capacity of our CONSISTI2V.

## Statement of Broader Impact

Conditional video synthesis aims at generating high-quality video with faithfulness to the given condition. It is a fundamental problem in computer vision and graphics, enabling diverse content creation and manipulation. Recent advances have shown great advances in generating aesthetical and high-resolution videos. However, the generated videos are still lacking coherence and consistency in terms of the subjects, background and style. Our work aims to address these issues and has shown promising improvement. However, our model also leads to slower motions in some cases. We believe this is still a long-standing issue that we would need to address before delivering it to the public.

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

# Appendix

## A   Additional Implementation Details

### A.1   Model Architecture

**U-Net Temporal Layers**   The temporal layers of our ConsistI2V share the same architecture as their spatial counterparts. For temporal convolution blocks, we create residual blocks containing two temporal convolution layers with a kernel size of $(3, 1, 1)$ along the temporal and spatial height and width dimensions. Our temporal attention blocks contain one temporal self-attention layer and one cross-attention layer that operates between temporal features and encoded text prompts. Following Blattmann et al. (2023), we also add a learnable weighing factor $\gamma$ in each temporal layer to combine the spatial and temporal outputs:

$$\mathbf{z}_{\texttt{out}} = \gamma \mathbf{z}_{\texttt{spatial}} + (1 - \gamma)\mathbf{z}_{\texttt{temporal}}, \gamma \in [0, 1], \tag{8}$$

where $\mathbf{z}_{\texttt{spatial}}$ denotes the output of the spatial layers (and thus the input to the temporal layers) and $\mathbf{z}_{\texttt{temporal}}$ represents the output of the temporal layers. We initialize all $\gamma = 1$ such that the temporal layers do not have any effects at the beginning of the training.

**Correlated Noise Initialization**   Existing I2V generation models (Xing et al., 2023; Zhang et al., 2023a; Zeng et al., 2023) often initialize the noise for each frame as i.i.d. Gaussian noises and ignore the correlation between consecutive frames. To effectively leverage the prior information that nearby frames in a video often share similar visual appearances, we employ the *mixed* noise prior from PYoCo (Ge et al., 2023) for noise initialization:

$$\epsilon_{\texttt{shared}} \sim \mathcal{N}(\mathbf{0}, \frac{\alpha^2}{1 + \alpha^2}\mathbf{I}), \epsilon_{\texttt{ind}}^i \sim \mathcal{N}(\mathbf{0}, \frac{1}{1 + \alpha^2}\mathbf{I}), \tag{9}$$

$$\epsilon^i = \epsilon_{\texttt{shared}} + \epsilon_{\texttt{ind}}^i, \tag{10}$$

where $\epsilon^i$ represents the noise of the $i^{\text{th}}$ frame, which consists of a shared noise $\epsilon_{\texttt{shared}}$ that has the same value across all frames and an independent noise $\epsilon_{\texttt{ind}}^i$ that is different for each frame. $\alpha$ controls the strength of the shared and the independent component of the noise and we empirically set $\alpha = 1.5$ in our experiments. We observed that this correlated noise initialization helps stabilize the training, prevents exploding gradients and leads to faster convergence.

**Positional Embeddings in Temporal Attention Layers**   We follow Wang et al. (2023d) and incorporate the rotary positional embeddings (RoPE) (Su et al., 2024) in the temporal attention layers of our model to indicate frame position information. To adapt RoPE embedding to our temporal first frame conditioning method as described in Section 3.4, given query vector of a certain frame at spatial position $(h, w)$, we rotate the key/value tokens in $\tilde{z}_h^1$ using the same angle as the first frame features at $(h, w)$ to indicate that this window of features also comes from the first frame.

**FPS Control**   We follow Xing et al. (2023) to use the sampling frame interval during training as a conditional signal to the model to enable FPS conditioning. Given a training video, we sample 16 frames by randomly choosing a frame interval $v$ between 1 and 5. We then input this frame interval value into the model by using the same method of encoding timestep embeddings: the integer frame interval value is first transformed into sinusoidal embeddings and then passed through two linear layers, resulting in a vector embedding that has the same dimension as the timestep embeddings. We then add the frame interval embedding and timestep embedding together and send the combined embedding to the U-Net blocks. We zero-initialize the second linear layer for the frame interval embeddings such that at the beginning of the training, the frame interval embedding is a zero vector.

### A.2   Training Paradigms

Existing I2V generation models (Xing et al., 2023; Zeng et al., 2023; Zhang et al., 2023a; 2024) often employ joint video-image training (Ho et al., 2022b) that trains the model on video-text and image-text data in

an interleaving fashion, or apply multi-stage training strategies that iteratively pretrain different model components using different types of data. Our model introduces two benefits over prior methods: (1) the explicit conditioning mechanism in the spatial and temporal self-attention layers effectively utilizes the visual cues from the first frame to render subsequent frames, thus reducing the difficulty of generating high-quality frames for the video diffusion model. (2) We directly employ the LDM VAE features as the conditional signal, avoiding training additional adaptor layers for other feature modalities (e.g. CLIP (Radford et al., 2021) image embeddings), which are often trained in separate stages by other methods. As a result, we train our model with a single video-text dataset in one stage, where we finetune all the parameters during training.

## B Model Evaluation Details

### B.1 Datasets

**UCF-101** (Soomro et al., 2012) is a human action recognition dataset consisting of 13K videos divided into 101 action categories. During the evaluation, we sample 2048 videos from the dataset based on the categorical distribution of the labels in the dataset. As the dataset only contains a label name for each category instead of descriptive captions, we employ the text prompts from PYoCo (Ge et al., 2023) for UCF-101 evaluation. The text prompts are listed below:

*applying eye makeup, applying lipstick, archery, baby crawling, gymnast performing on a balance beam, band marching, baseball pitcher throwing baseball, a basketball player shooting basketball, dunking basketball in a basketball match, bench press, biking, billiards, blow dry hair, blowing candles, body weight squats, a person bowling on bowling alley, boxing punching bag, boxing speed bag, swimmer doing breast stroke, brushing teeth, clean and jerk, cliff diving, bowling in cricket gameplay, batting in cricket gameplay, cutting in kitchen, diver diving into a swimming pool from a springboard, drumming, two fencers have fencing match indoors, field hockey match, gymnast performing on the floor, group of people playing frisbee on the playground, swimmer doing front crawl, golfer swings and strikes the ball, haircuting, a person hammering a nail, an athlete performing the hammer throw, an athlete doing handstand push up, an athlete doing handstand walking, massagist doing head massage to man, an athlete doing high jump, horse race, person riding a horse, a woman doing hula hoop, ice dancing, athlete practicing javelin throw, a person juggling with balls, a young person doing jumping jacks, a person skipping with jump rope, a person kayaking in rapid water, knitting, an athlete doing long jump, a person doing lunges with barbell, military parade, mixing in the kitchen, mopping floor, a person practicing nunchuck, gymnast performing on parallel bars, a person tossing pizza dough, a musician playing the cello in a room, a musician playing the daf, a musician playing the indian dhol, a musician playing the flute, a musician playing the guitar, a musician playing the piano, a musician playing the sitar, a musician playing the tabla, a musician playing the violin, an athlete jumps over the bar, gymnast performing pommel horse exercise, a person doing pull ups on bar, boxing match, push ups, group of people rafting on fast moving river, rock climbing indoor, rope climbing, several people rowing a boat on the river, couple salsa dancing, young man shaving beard with razor, an athlete practicing shot put throw, a teenager skateboarding, skier skiing down, jet ski on the water, sky diving, soccer player juggling football, soccer player doing penalty kick in a soccer match, gymnast performing on still rings, sumo wrestling, surfing, kids swing at the park, a person playing table tennis, a person doing TaiChi, a person playing tennis, an athlete practicing discus throw, trampoline jumping, typing on computer keyboard, a gymnast performing on the uneven bars, people playing volleyball, walking with dog, a person doing pushups on the wall, a person writing on the blackboard, a kid playing Yo-Yo.*

**MSR-VTT** (Xu et al., 2016) is an open-domain video retrieval and captioning dataset containing 10K videos, with 20 captions for each video. The standard splits for MSR-VTT include 6,513 training videos, 497 validation videos and 2,990 test videos. We use the official test split in the experiment and randomly select a text prompt for each video during evaluation.

## B.2 Evaluation Metrics

**Fréchet Video Distance (FVD)**  FVD measures the similarity between generated and real videos. We follow Blattmann et al. (2023) to use a pretrained I3D model[3] to extract features from the videos and use the official UCF FVD evaluation code[4] from Ge et al. (2022) to compute FVD statistics.

**Fréchet Inception Distance (FID)**  We use a pretrained Inception model[5] to extract per-frame features from the generated and real videos to compute FID. Our evaluation code is similar to the evaluation script[6] provided by Brooks et al. (2022).

**Inception Score (IS)**  We compute a video version of the Inception Score following previous works (Blattmann et al., 2023; Saito et al., 2020), where the video features used in IS are computed from a C3D model (Tran et al., 2015) pretraind on UCF-101. We use the TorchScript C3D model[7] and employ the evaluation code[8] from (Skorokhodov et al., 2022).

**CLIP Similarity (CLIPSIM)**  Our CLIPSIM metrics are computed using TorchMetrics. We use the CLIP-VIT-B/32 model (Radford et al., 2021) to compute the CLIP similarity for all frames in the generated videos and report the averaged results.

## C  I2V-Bench Evaluation Metrics

We make use of the metrics provided by VBench (Huang et al., 2023b) in the I2V-Bench evaluation process.

**Background Consistency**  We measure the temporal consistency of the background scenes by computing the CLIP (Radford et al., 2021) feature similarity across frames. CLIP features represent the high-level semantic content of an image and are not sensitive to subject variations, making it a suitable feature to measure background consistency.

**Subject Consistency**  We measure the subject consistency by calculating the DINO (Caron et al., 2021) feature similarity across frames. Since DINO features capture fine-grained semantic information in the foreground subjects, they are suitable for evaluating subject consistency.

**Temporal Flickering**  We calculate the average absolute difference between each frame. Temporal flickerings in generated videos will cause large local pixel value variations, thereby causing large frame differences.

**Motion Smoothness**  We adapt AMT (Li et al., 2023) to evaluate the level of smoothness in the generated motion. Specifically, given a generated video, we drop the odd-number frames to obtain a video with a lower frame rate. We then use AMT to interpolate the video back to the original frame rate. Motion smoothness is defined as the mean absolute error (MAE) between the reconstructed frames and the dropped frames. A large MAE indicates that the dropped frames contain abrupt motions that are not likely to be interpolated by AMT, thus indicating a lack of smooth motion progression in the generated video.

**Dynamic Degree**  We adopt RAFT (Teed & Deng, 2020) to estimate the degree of dynamics in the video. RAFT is a method designed for optical flow estimation. We use the magnitude of the flow vector to indicate the degree of dynamics presented in the video.

---

[3]https://github.com/songweige/TATS/blob/main/tats/fvd/i3d_pretrained_400.pt
[4]https://github.com/pfnet-research/tgan2
[5]https://api.ngc.nvidia.com/v2/models/nvidia/research/stylegan3/versions/1/files/metrics/inception-2015-12-05.pkl
[6]https://github.com/NVlabs/long-video-gan/tree/main
[7]https://www.dropbox.com/s/jxpu7avzdc9n97q/c3d_ucf101.pt?dl=1
[8]https://github.com/universome/stylegan-v/tree/master

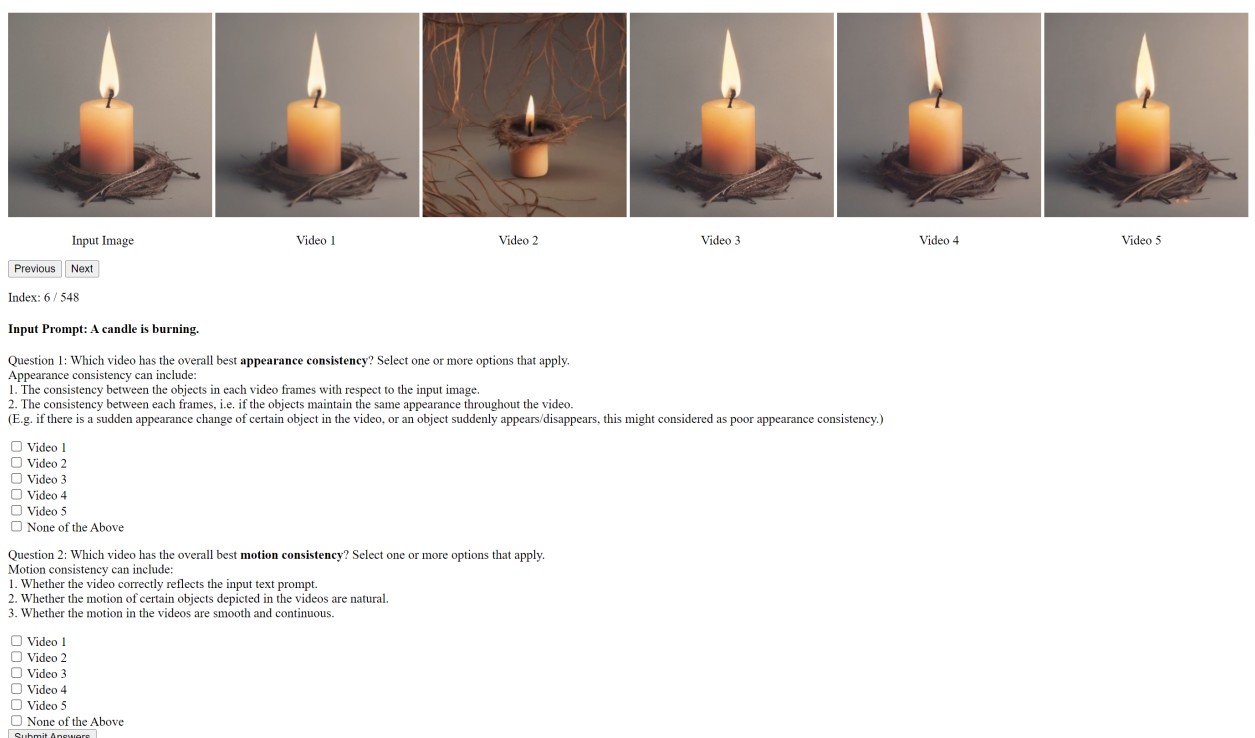

Figure 9: Interface of our human evaluation experiment.

**Object Consistency** We medially select 100 videos from the Pet, Vehicle, Animal, and Food categories in the I2V-Bench validation dataset. We then annotate each video based on the objects presented in the video (e.g. "cat", "car", etc.) and use GRiT (Wu et al., 2022) to determine if the generated video reflects the annotated objects. Object consistency is computed as the success rate of synthesizing the annotated objects in the generated videos.

**Scene Consistency** We select 100 videos from the Scenery-Nature and Scenery-City categories in the I2V-Bench validation dataset. We then annotate the scenes in the videos as different keywords (e.g. "ocean", "sky"). Scene consistency is computed by applying Tag2Text (Huang et al., 2023a) to caption the generated videos and detect whether the captions in the generated videos contain the annotated keywords.

**Overall Video-text Consistency** We adopt the overall video-text consistency by computing the cosine similarity between video and text embeddings using ViCLIP (Wang et al., 2023e) to reflect the semantics consistency of the manually annotated captions of I2V-Bench and the generated videos. ViCLIP is a CLIP-based model trained on video-text data that maps videos and text in a joint embedding space. Therefore, this similarity represents an overall degree of alignment between the generated video and the text prompt.

# D Human Evaluation Details

We show our designed human evaluation interface in Figure 9. We collect 274 prompts and use Pixart-$\alpha$ (Chen et al., 2023b) and SDXL (Podell et al., 2023) to generate 548 images as the input first frame. We then generate a video for each input image using the four baseline models I2VGen-XL (Zhang et al., 2023a), DynamiCrafter (Xing et al., 2023), SEINE (Chen et al., 2023c) and AnimateAnything (Dai et al., 2023), as well as our ConsistI2V. To ensure a fair comparison, we resize and crop all the generated videos to $256 \times 256$ and truncate all videos to 16 frames, 2 seconds (8 FPS). We then randomly shuffle the order of these 548 samples for each participant and let them answer the two questions regarding rating the appearance and motion consistency of the videos for a subset of samples.

## E  I2V-Bench Statistics

Table 4: I2V-Bench Statistics

| Category | Count | Category | Count |
|---|---|---|---|
| Portrait | 663 | Animation-Static | 120 |
| Scenery-Nature | 500 | Music | 63 |
| Pet | 393 | Game | 61 |
| Food | 269 | Animal | 55 |
| Animation-Hard | 187 | Industry | 44 |
| Science | 180 | Painting | 40 |
| Sports | 149 | Others | 40 |
| Scenery-City | 138 | Vehicle | 30 |
| Drama | 19 | **Total** | **2951** |

## F  Additional Quantitative Results

Table 5: Experimental results for T2V generation on MSR-VTT.

| Method | #data | #params | CLIPSIM ($\uparrow$) | FVD ($\downarrow$) |
|---|---|---|---|---|
| CogVideo (EN) (Hong et al., 2022) | 5.4M | 15.5B | 0.2631 | 1294 |
| MagicVideo (Zhou et al., 2022) | 10M | - | - | 1290 |
| LVDM (He et al., 2022) | 2M | 1.2B | 0.2381 | 742 |
| VideoLDM (Blattmann et al., 2023) | 10M | 4.2B | 0.2929 | - |
| InternVid (Wang et al., 2023e) | 28M | - | 0.2951 | - |
| ModelScope (Wang et al., 2023a) | 10M | 1.7B | 0.2939 | 550 |
| Make-A-Video (Singer et al., 2022) | 20M | 9.7B | 0.3049 | - |
| Latent-Shift (An et al., 2023) | 10M | 1.5B | 0.2773 | - |
| VideoFactory (Wang et al., 2023b) | - | 2.0B | 0.3005 | - |
| PixelDance (Zeng et al., 2023) | 10M | 1.5B | 0.3125 | 381 |
| ConsistI2V | 10M | 1.7B | 0.2968 | 428 |

To compare against more closed-sourced I2V methods and previous T2V methods, we conduct an additional quantitative experiment following PixelDance (Zeng et al., 2023) and evaluate our model's ability as a generic T2V generator by using Stable Diffusion 2.1-base (Rombach et al., 2022) to generate the first frame conditioned on the input text prompt. We employ the test split of MSR-VTT (Xu et al., 2016) and evaluate FVD and CLIPSIM for this experiment. As shown in Table 5, our method is on par with previous art in T2V generation and achieves a second-best FVD result of 428 and a comparable CLIPSIM of 0.2968. These results indicate our model's capability of handling diverse video generation tasks.

## G  I2V-Bench Results

We present the detailed breakdown of I2V-Bench results in Table 6, 7, 8, 9, 10, 11, 12 and 13.

Table 6: Results of I2V-Bench for Background Consistency. S-N, S-C, A-H and A-S respectively represent Scenery-Nature, Scenery-City, Animation-Hard and Animation-Static.

| Category | I2VGen-XL | AnimateAnything | DynamiCrafter | SEINE | ConsistI2V |
|---|---|---|---|---|---|
| Portrait | 94.04 | 98.42 | 92.61 | 94.07 | 93.38 |
| S-N | 95.26 | 98.64 | 93.74 | 95.45 | 95.60 |
| Pet | 91.78 | 97.92 | 92.32 | 91.59 | 94.26 |
| Food | 97.28 | 98.97 | 96.13 | 96.89 | 96.97 |
| A-H | 94.32 | 98.68 | 93.63 | 89.03 | 93.29 |
| Science | 95.65 | 98.67 | 92.63 | 94.44 | 95.20 |
| Sports | 92.98 | 98.88 | 93.11 | 94.93 | 94.26 |
| S-C | 94.02 | 98.47 | 93.05 | 94.26 | 94.50 |
| A-S | 96.83 | 98.82 | 95.10 | 95.39 | 96.43 |
| Music | 93.20 | 97.62 | 93.12 | 93.34 | 94.33 |
| Game | 92.52 | 97.66 | 92.81 | 88.34 | 93.20 |
| Animal | 96.64 | 98.14 | 95.64 | 96.14 | 95.86 |
| Industry | 97.36 | 98.93 | 95.81 | 97.26 | 96.27 |
| Painting | 95.31 | 98.36 | 93.18 | 91.82 | 94.87 |
| Others | 95.59 | 98.86 | 94.96 | 94.74 | 93.75 |
| Vehicle | 95.16 | 98.77 | 94.83 | 95.50 | 96.23 |
| Drama | 94.32 | 98.90 | 91.63 | 92.97 | 93.95 |
| All | 94.52 | 98.50 | 93.47 | 93.98 | 94.69 |

Table 7: Results of I2V-Bench for Subject Consistency. S-N, S-C, A-H and A-S respectively represent Scenery-Nature, Scenery-City, Animation-Hard and Animation-Static.

| Category | I2VGen-XL | AnimateAnything | DynamiCrafter | SEINE | ConsistI2V |
|---|---|---|---|---|---|
| Portrait | 89.97 | 97.92 | 87.50 | 89.00 | 89.07 |
| S-N | 90.94 | 98.06 | 88.72 | 91.34 | 92.65 |
| Pet | 80.92 | 97.05 | 83.80 | 80.79 | 89.11 |
| Food | 95.19 | 98.46 | 92.93 | 94.27 | 95.03 |
| A-H | 86.28 | 97.67 | 87.19 | 78.55 | 86.34 |
| Science | 90.97 | 98.31 | 85.62 | 87.64 | 91.29 |
| Sports | 89.22 | 98.91 | 89.63 | 93.59 | 92.57 |
| S-C | 89.62 | 98.12 | 88.36 | 90.85 | 91.30 |
| A-S | 92.57 | 97.97 | 89.12 | 90.06 | 92.64 |
| Music | 88.99 | 95.86 | 90.06 | 90.12 | 93.17 |
| Game | 86.60 | 96.78 | 87.25 | 78.14 | 88.61 |
| Animal | 93.27 | 97.21 | 90.74 | 91.08 | 91.82 |
| Industry | 95.84 | 99.05 | 91.42 | 95.56 | 94.37 |
| Painting | 89.62 | 98.15 | 86.97 | 81.15 | 91.25 |
| Others | 90.98 | 98.27 | 89.88 | 87.03 | 86.23 |
| Vehicle | 90.08 | 98.28 | 91.38 | 90.36 | 92.86 |
| Drama | 91.09 | 98.65 | 85.56 | 87.57 | 89.62 |
| All | 89.36 | 97.90 | 88.01 | 88.21 | 90.85 |

Table 8: Results of I2V-Bench for Temporal Flickering. S-N, S-C, A-H and A-S respectively represent Scenery-Nature, Scenery-City, Animation-Hard and Animation-Static.

| Category | I2VGen-XL | AnimateAnything | DynamiCrafter | SEINE | ConsistI2V |
|---|---|---|---|---|---|
| Portrait | 94.89 | 99.25 | 93.85 | 96.78 | 97.17 |
| S-N | 94.58 | 99.07 | 94.54 | 96.67 | 96.86 |
| Pet | 90.98 | 98.90 | 92.24 | 93.89 | 95.88 |
| Food | 95.39 | 98.74 | 95.36 | 96.45 | 96.12 |
| A-H | 93.22 | 99.26 | 93.90 | 95.11 | 96.18 |
| Science | 94.71 | 99.41 | 91.78 | 96.79 | 97.56 |
| Sports | 91.71 | 98.97 | 92.66 | 95.37 | 95.34 |
| S-C | 93.16 | 99.00 | 93.82 | 95.89 | 96.69 |
| A-S | 95.09 | 98.90 | 94.45 | 96.19 | 96.63 |
| Music | 92.78 | 99.15 | 93.80 | 95.96 | 96.05 |
| Game | 93.87 | 99.26 | 93.24 | 96.46 | 97.88 |
| Animal | 96.66 | 98.83 | 96.56 | 97.24 | 96.51 |
| Industry | 96.53 | 99.13 | 95.39 | 97.35 | 96.90 |
| Painting | 95.36 | 99.12 | 94.11 | 95.89 | 97.07 |
| Others | 96.23 | 99.39 | 94.97 | 96.96 | 97.84 |
| Vehicle | 95.23 | 99.28 | 95.51 | 96.60 | 97.25 |
| Drama | 94.48 | 98.92 | 91.70 | 95.81 | 95.88 |
| All | 94.03 | 99.08 | 93.81 | 96.08 | 96.65 |

Table 9: Results of I2V-Bench for Motion Smoothness. S-N, S-C, A-H and A-S respectively represent Scenery-Nature, Scenery-City, Animation-Hard and Animation-Static.

| Category | I2VGen-XL | AnimateAnything | DynamiCrafter | SEINE | ConsistI2V |
|---|---|---|---|---|---|
| Portrait | 96.73 | 99.36 | 95.83 | 98.31 | 98.17 |
| S-N | 96.44 | 99.22 | 96.49 | 98.14 | 97.84 |
| Pet | 93.54 | 99.09 | 94.48 | 97.13 | 97.39 |
| Food | 97.30 | 98.97 | 96.97 | 98.03 | 97.28 |
| A-H | 95.03 | 99.34 | 95.88 | 97.06 | 97.54 |
| Science | 96.42 | 99.49 | 95.31 | 98.25 | 98.37 |
| Sports | 95.07 | 99.15 | 95.62 | 97.91 | 97.23 |
| S-C | 95.30 | 99.18 | 95.86 | 97.89 | 97.64 |
| A-S | 96.66 | 99.09 | 96.01 | 97.68 | 97.77 |
| Music | 95.20 | 99.27 | 95.96 | 98.22 | 97.21 |
| Game | 96.00 | 99.35 | 95.56 | 97.82 | 98.59 |
| Animal | 97.92 | 99.04 | 97.53 | 98.22 | 97.48 |
| Industry | 97.96 | 99.25 | 96.81 | 98.45 | 97.84 |
| Painting | 96.58 | 99.24 | 95.60 | 97.28 | 97.77 |
| Others | 97.46 | 99.47 | 96.68 | 98.34 | 98.50 |
| Vehicle | 96.91 | 99.37 | 96.95 | 98.21 | 98.20 |
| Drama | 97.15 | 99.08 | 94.82 | 97.88 | 97.11 |
| All | 96.03 | 99.23 | 95.89 | 97.92 | 97.77 |

Table 10: Results of I2V-Bench for object consistency.

| Category | I2VGen-XL | AnimateAnything | DynamiCrafter | SEINE | ConsistI2V |
|---|---|---|---|---|---|
| Pet | 65.00 | 73.50 | 71.75 | 82.25 | 76.00 |
| Food | 0.00 | 0.25 | 0.00 | 0.00 | 1.00 |
| Animal | 25.00 | 28.00 | 24.25 | 28.00 | 25.50 |
| Vehicle | 27.00 | 28.50 | 32.75 | 31.50 | 25.75 |
| All | 29.25 | 32.56 | 32.18 | 35.43 | 32.06 |

Table 11: Results of I2V-Bench for scene consistency. S-N and S-C respectively represent Scenery-Nature and Scenery-City

| Category | I2VGen-XL | AnimateAnything | DynamiCrafter | SEINE | ConsistI2V |
|---|---|---|---|---|---|
| S-N | 33.70 | 18.49 | 34.80 | 37.50 | 33.45 |
| S-C | 12.88 | 35.53 | 12.62 | 17.47 | 15.81 |
| All | 23.50 | 27.18 | 23.93 | 27.68 | 24.81 |

Table 12: Results of I2V-Bench for Dynamic Degree. S-N, S-C, A-H and A-S respectively represent Scenery-Nature, Scenery-City, Animation-Hard and Animation-Static.

| Category | I2VGen-XL | AnimateAnything | DynamiCrafter | SEINE | ConsistI2V |
|---|---|---|---|---|---|
| Portrait | 57.01 | 3.47 | 62.90 | 35.44 | 35.14 |
| S-N | 50.60 | 2.00 | 56.80 | 27.20 | 26.80 |
| Pet | 92.88 | 6.62 | 92.62 | 85.24 | 61.32 |
| Food | 36.06 | 6.32 | 50.93 | 29.37 | 28.25 |
| A-H | 63.64 | 6.42 | 60.96 | 63.10 | 56.15 |
| Science | 47.78 | 0.00 | 69.44 | 37.78 | 34.44 |
| Sports | 87.25 | 2.01 | 93.29 | 46.98 | 57.05 |
| S-C | 50.00 | 2.17 | 55.07 | 29.71 | 23.91 |
| A-S | 28.33 | 0.00 | 37.50 | 26.67 | 20.83 |
| Music | 65.08 | 9.52 | 69.84 | 38.10 | 46.03 |
| Game | 70.49 | 8.20 | 67.21 | 49.18 | 24.59 |
| Animal | 40.00 | 3.64 | 34.55 | 23.64 | 27.27 |
| Industry | 22.73 | 0.00 | 47.73 | 15.91 | 22.73 |
| Painting | 52.50 | 2.50 | 57.50 | 45.00 | 37.50 |
| Others | 32.50 | 2.50 | 40.00 | 35.00 | 17.50 |
| Vehicle | 53.33 | 0.00 | 50.00 | 43.33 | 43.33 |
| Drama | 57.89 | 0.00 | 63.16 | 52.63 | 42.11 |
| All | 57.88 | 3.69 | 64.11 | 42.12 | 37.48 |

Table 13: Results of I2V-Bench for Text-Video Overall Consistency. S-N, S-C, A-H and A-S respectively represent Scenery-Nature, Scenery-City, Animation-Hard and Animation-Static.

| Category | I2VGen-XL | AnimateAnything | DynamiCrafter | SEINE | ConsistI2V |
|---|---|---|---|---|---|
| Portrait | 15.50 | 17.68 | 17.95 | 19.27 | 18.64 |
| S-N | 17.51 | 18.51 | 18.12 | 19.70 | 19.14 |
| Pet | 20.23 | 23.10 | 22.94 | 24.12 | 23.29 |
| Food | 19.80 | 21.26 | 21.35 | 21.98 | 21.57 |
| A-H | 11.23 | 12.82 | 12.79 | 17.61 | 15.26 |
| Science | 17.26 | 18.80 | 18.48 | 19.60 | 19.59 |
| Sports | 20.11 | 21.92 | 22.52 | 22.59 | 21.69 |
| S-C | 15.85 | 17.16 | 17.21 | 18.38 | 18.23 |
| A-S | 13.63 | 14.91 | 15.32 | 16.20 | 15.49 |
| Music | 18.69 | 21.89 | 21.63 | 23.71 | 24.14 |
| Game | 12.63 | 15.35 | 13.27 | 18.53 | 15.63 |
| Animal | 18.20 | 19.50 | 19.06 | 19.78 | 19.62 |
| Industry | 17.38 | 19.51 | 19.55 | 19.81 | 19.90 |
| Painting | 10.66 | 13.72 | 13.26 | 17.67 | 15.48 |
| Others | 14.66 | 17.63 | 18.01 | 19.13 | 19.64 |
| Vehicle | 17.98 | 19.00 | 18.97 | 19.53 | 19.08 |
| Drama | 12.01 | 13.13 | 11.52 | 16.46 | 14.81 |
| All | 16.89 | 18.74 | 18.68 | 20.21 | 19.50 |

## H   Additional I2V Generation Results

We showcase more I2V generation results for ConsistI2V in Figure 10 and Figure 11.

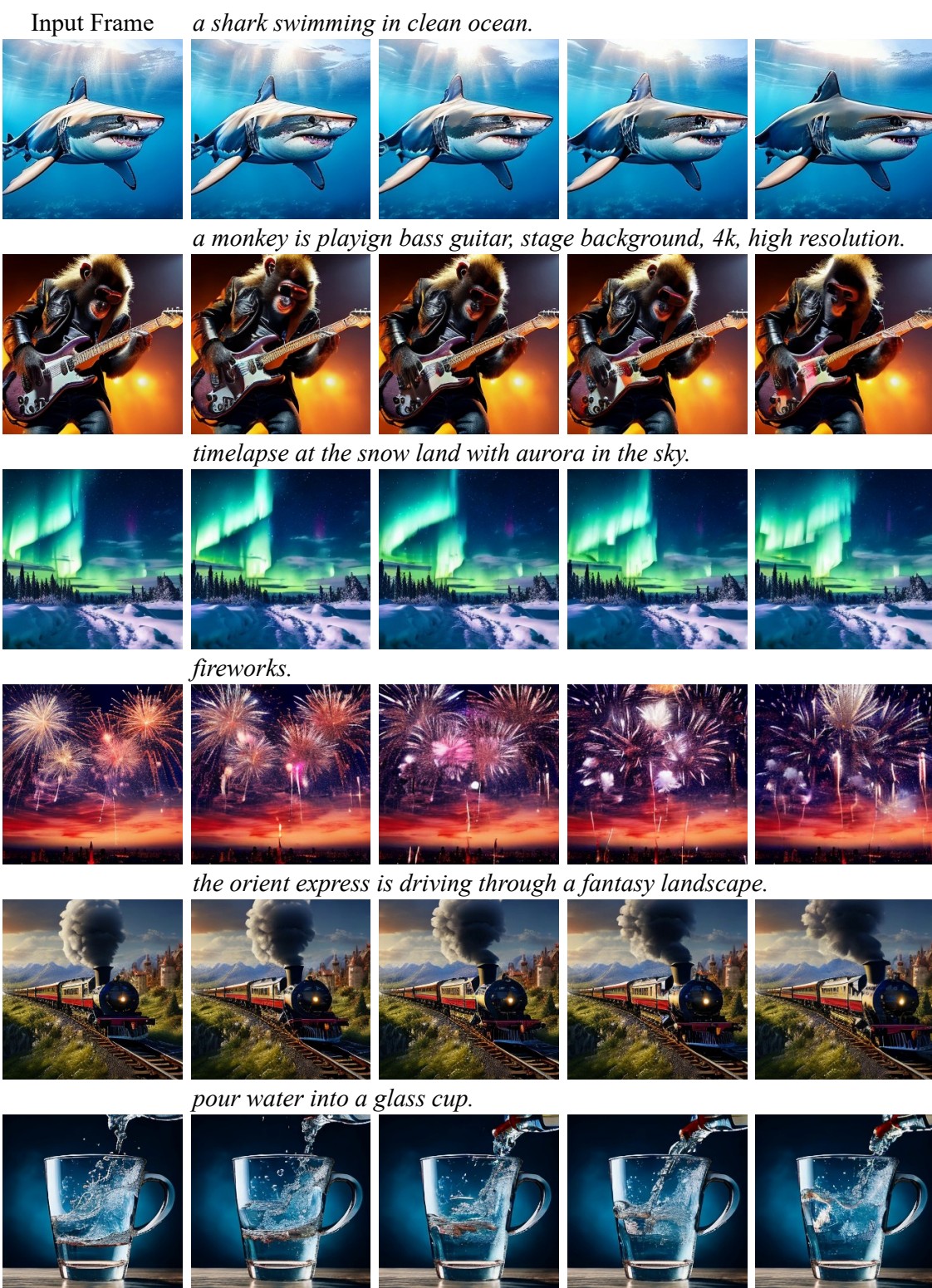

Figure 10: Additional I2V generation results for CONSISTI2V.

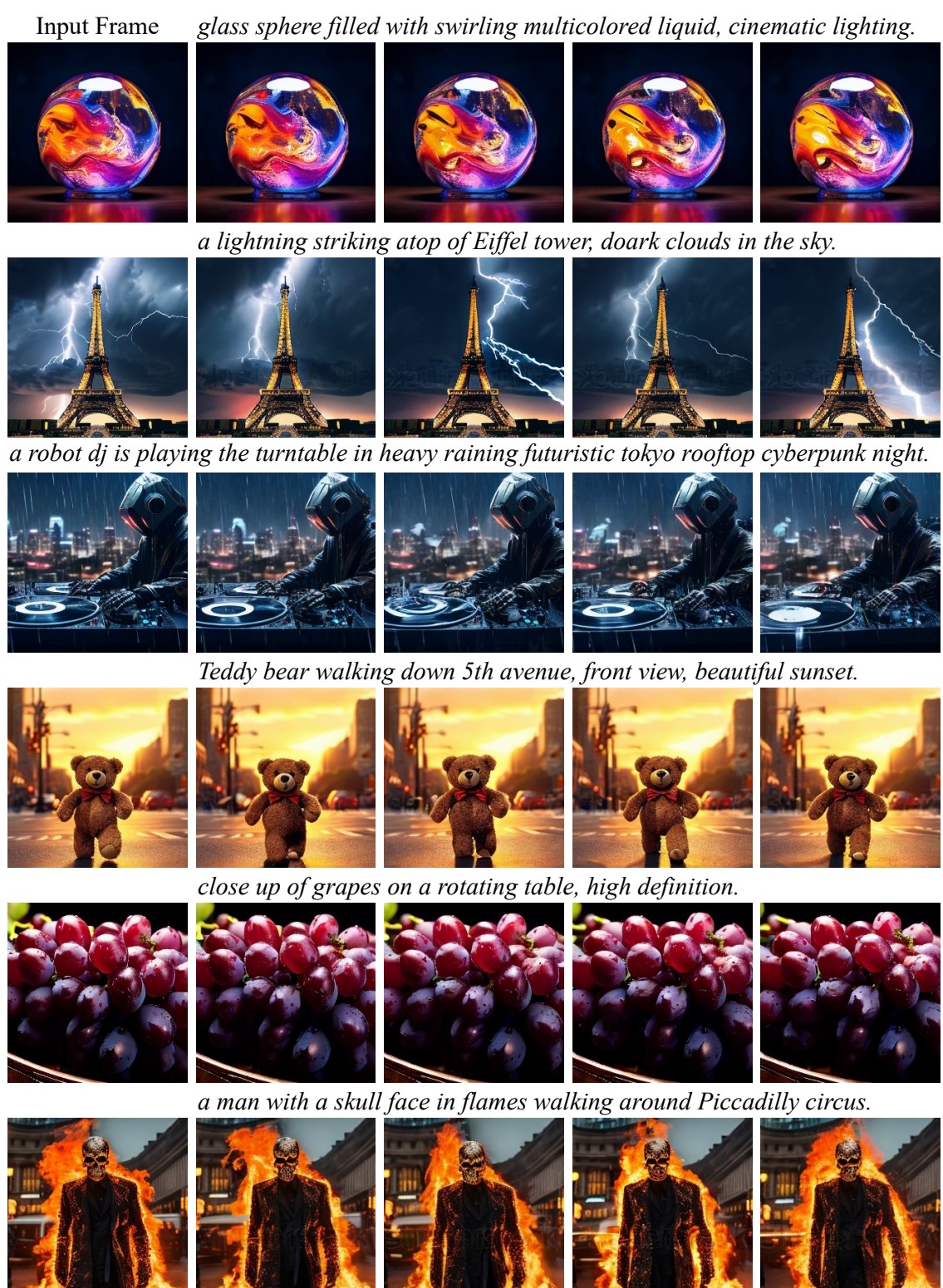

Figure 11: Additional I2V generation results for CONSISTI2V.

