# OpenReview forum: "ConsistI2V: Enhancing Visual Consistency for Image-to-Video Generation"
_TMLR — Accepted by TMLR_

### Review · Reviewer_ashH · 2024-04-20

**Summary Of Contributions:**

This paper proposes an image-to-video generation model that predicts future frames based on an initial frame and text input. The authors focus on enhancing the consistency between the generated video and the given initial frame. To achieve this, the paper introduces a spatial conditioning mechanism that conditions every layer on the first frame and an augmented temporal self-attention that allows the temporal layer to observe the surroundings of the first frame. Finally, a novel noise initialization technique is proposed to effectively increase the temporal consistency of the generated video. The supplementary materials, which include the baseline and the results generated by the proposed methodology, demonstrate the superiority of the proposed approach.

**Audience:**

No

**Broader Impact Concerns:**

There are no broader impact concerns.

**Claims And Evidence:**

Yes

**Requested Changes:**

1.	Figure 5 needs to adequately represent the advantages of the proposed method by increasing the frame stride to include diverse frames. Judging solely by the figure, the first frame appears static, similar to AnimateAnyone.

**Strengths And Weaknesses:**

Strengths:

1.	The proposed spatial conditioning method and FrameInit are novel. Specifically, the idea of FrameInit is reasonable and can be applied to other image-to-video method to enhance consistency.

2.	The videos provided in the supplementary materials clearly demonstrate the superior generation quality of the proposed method.

3.	The proposed method quantitively outperforms the existing baselines while utilizing a smaller amount of a training dataset.

Weaknesses & Questions:

1.	The proposed method adopts a single-stage training strategy. The authors argue that a problem arises when using 1D temporal self-attention as the temporal layer, in which the temporal self-attention only observes patches at the same spatial position. This might be suitable for a two-stage training strategy where spatial and temporal layers are trained separately but appears to be less effective in this paper, where spatial convolution layers and temporal self-attention layers are trained simultaneously. Consequently, as shown in Table 3 in the main paper, there is no notable quantitative difference.

2.	It would be beneficial to analyze the effect of the hyper-parameter in FrameInit. Specifically, on the potential trade-off between consistency and diversity in the generated videos, controlled by the hyper-parameters of FrameInit.

3.	A detailed explanation for the automatic evaluation in section 5.2 would be helpful for a clear understanding, as the experimental setup and their implications in Table 2 seem to be ambiguous.

4.	Other baselines, such as SEINE and I2VGEN-XL, use more training data. What is the reason for not following their training protocols?

---

> ### Author Response · Authors · 2024-04-26
> **Response to Reviewer ashH**
>
> We thank the reviewer for recognizing the strength and novelty of our method and raising constructive and valuable feedback. We would like to respond to the weaknesses and questions in our paper as follows:
>
> **Responses to Weaknesses & Questions:**
> 1. We have done experiments related to training the spatial and temporal layers in multiple stages but didn’t observe performance improvements compared to the current single-stage training scheme. Previous methods either left the spatial layers untouched (e.g. SEINE) or injected semantic image embeddings (e.g. CLIP image embeddings) into the spatial layers (e.g. DynamiCrafter). As a result, these methods benefit from finetuning the image diffusion backbone on additional image data before video training. In contrast, we employ cross-frame attention mechanisms in the spatial layers. Consequently, the spatial layers of our model should also be trained on video data, and we find that training the spatial and temporal layers simultaneously yields the best model.
>
> 2. In general, we find that decreasing $\tau$ (number of steps of adding the inference noise to the static video latent) and increasing $D_0$ (spatiotemporal stop frequency for the Gaussian low-pass filter) both result in the generated video having less subject appearance variation and abrupt motion, leading to better visual consistency. However, as the conditional signal from the static video layout becomes stronger, the generated videos also display smaller motion magnitude and the model sometimes may produce almost static videos. We select $\tau=850$ and $D_0=0.25$ in the video generation process to strike a balance between video consistency and motion diversity. For camera motion control, as it requires stronger layout guidance to ensure the generated video follows the synthetic camera motion, we consequently decrease $\tau$ to 750 and increase $D_0$ to 0.5. We will include a detailed analysis of these hyperparameters in the revision.
> 3. We have included a brief summary of the metrics that we used in the I2V-Bench evaluation in Appendix C. We will add more discussions on the implementation of these metrics and their implications in the revision.
> 4. We note that SEINE and I2VGen-XL obtained more training data from their internal private datasets. On the other hand, we focus on training and evaluating our model entirely on public datasets to ensure reproducibility. Moreover, as DynamiCrafter and AnimateAnything are also mainly trained on WebVid-10M, we choose this dataset to avoid unfair comparison against these baseline methods. We envision that training on a larger dataset will likely further increase the performance of our model and leave this as a future work.
>
> **Response to Requested Changes:** Thanks for pointing out this issue. We noticed that the motion progression of some videos is less recognizable in the figure when the video is broken down into individual frames. We will select better examples to emphasize the advantages of our method in the revision.
>
> We hope the above information helps resolve your concerns and we are willing to clarify any further questions. Thanks again for your valuable feedback!

---

### Review · Reviewer_hVB1 · 2024-04-25

**Summary Of Contributions:**

The paper proposes a diffusion generative video model using a single initialization frame and a text prompt as a conditioning signal. The main contributions are the proposed spatiotemporal conditioning mechanisms in the attention blocks in the denoiser architecture and an empirical scheme for initializing the latents during inference (a.k.a FrameInit). The authors also present a benchmark I2V-Bench for evaluating the proposed method.

**Audience:**

Yes

**Broader Impact Concerns:**

These concerns have been addressed adequately.

**Claims And Evidence:**

Yes

**Requested Changes:**

See suggestions/questions above

**Strengths And Weaknesses:**

Strengths:
1. The proposed method is simple and feels intuitive.
2. The experiments demonstrate the advantage of the proposed method over existing baselines, both quantitatively and qualitatively. In general, the qualitative results look appealing.

Questions/Suggestions:

1. What is the intuition behind combining the spatial and temporal information in the UNet in Eqn. 8 in Appendix 1?
2. It would help to formally specify the additional computational costs incurred for the self-attention operation with the concatenation proposed in Eqn. 3 and Eqn. 4 in contrast to traditional self-attention mechanisms in video diffusion models.
3. The intuition behind the concatenation in Eqn. 4 is clear to me. However, is it correct to say that introducing a window of size K introduces a trade-off between consistency in long-video generation and the cost of additional self-attention computations? For instance, with a value of K=3, the authors use only a very local context window, so consistency in the generated frames with respect to the first frame would be affected. It would be great if the authors could elaborate more on this aspect in the main text for more clarity to the reader.
4. The ablation results in Table 3 are interesting. Is there any intuition on why removing temporal conditioning in the attention operation actually improves the evaluation metrics? From the presentation in Eqn. 4, it makes sense intuitively, but the empirical results state otherwise. Could it be because of a small K value? I think the presentation could be improved if the authors could share any additional qualitative or quantitative results for this ablation. Fig. 6 illustrates the importance of temporal conditioning, but this is just one example.
5. Limitations: From the standpoint of methodological or empirical experiments, I would highly encourage the authors to add limitations of their work in the paper.

Minor points:
1. Table 1: FID scores for ConsistI2V are highlighted in bold while AnimateAnything is the best (15.74 vs 10.0). In general, the tables could be simplified by highlighting the best results by not omitting specific methods.
2. Under the section Quantitative evaluation/I2V-Bench, I believe Table 5 should be Table 2?

---

> ### Author Response · Authors · 2024-05-02
> **Response to Reviewer hVB1**
>
> We thank the reviewer for appreciating the strength and effectiveness of our method and for providing valuable feedback. Here is our response to the weaknesses and questions raised in our paper:
>
> **Responses to Questions/Suggestions:**
> 1. The intuition behind mixing the spatial and temporal outputs after every temporal layer in the UNet is to ensure the pretrained weights from the base image diffusion backbone are effectively leveraged. At the beginning of the training, we set $\gamma$ (weighing factor between spatial and temporal output) to 1 such that the temporal layers are disabled. As the spatial layers are initialized from the pretrained Stable Diffusion, the model can still generate legitimate frames, although they are not temporally correlated. During training, the $\gamma$ parameter at each temporal layer can be progressively updated toward including more information from the temporal layers, such that the frames become more aligned in the temporal dimension and eventually form a coherent video.
>
> 2. Please refer to the table below for detailed inference cost statistics. All models are measured on a single Nvidia RTX 4090 with float32 and an inference batch size of 1. In general, we find that the proposed spatiotemporal self-attention operations and FrameInit introduce limited computational overhead during inference (~200Mb extra GPU memory and ~1.3s additional inference time). We will also update this table in the revision.
>
> |                                   | GPU Memory (Mb) | Inference Runtime (s) | TFLOPs (UNet) |
> |-----------------------------------|:---------------:|:---------------------:|:-------------:|
> | ConsistI2V                        |       9249      |         18.48         |     5.168     |
> | w/o FrameInit                     |       9245      |         18.37         |     5.168     |
> | w/o FrameInit & T.Cond.           |       9233      |         17.93         |     5.117     |
> | w/o FrameInit & T.Cond. & S.Cond. |       9017      |         17.14         |     5.015     |
>
> 3. There indeed exists a trade-off between the granularity of the temporal feature conditioning and the cost of the additional self-attention computations. Different window sizes will likely affect the consistency of the generated frames to the first frame. However, we also note that as the UNet downsamples the input latent in its intermediate layers, and the input latent is already downsampled 8x by the VAE, even a window size of K=3 can still create a much larger receptive field in deeper UNet layers. Furthermore, as mentioned in Section 3.4, previous methods adopt temporal attention that employs an even smaller context window (K=1). For example, in the second last level of the UNet (4x downsampling compared to the input latent), a $256\times 256$ video will be compressed into a $8\times 8$ feature map. Consequently, a $3 \times 3$ window on this feature map covers a region of $96 \times 96$ in the input first frame. This is a significant increase from previous methods where a $1 \times 1$ window on the feature map corresponded to only a $24 \times 24$ region in the original frame. We will include more discussions on this aspect in the revision.
>
> 4. We would like to point out that while UCF-101 is being widely used as one of the standard evaluation benchmarks for video generation, the quantitative results on UCF-101 sometimes cannot properly reflect the quality of the generated video. This can be illustrated in the relatively good UCF results for AnimateAnything in Table 1 and has also been discussed in other video generation works such as EMU-Video [1] (see Table 2 in [1]). Therefore, we believe a small drop in UCF scores does not necessarily indicate the model generates worse-quality videos after applying temporal conditioning. Furthermore, our goal of applying temporal conditioning is to improve the visual consistency of the generated video frames with respect to the first frame. On the other hand, the automatic metrics in the UCF benchmark mainly focus on assessing the video’s overall quality (IS) and similarity to the real UCF videos (FVD), as well as the quality of the video frames (FID). These metrics may not be able to reflect the improved frame consistency introduced by the temporal conditioning mechanism. We will include more qualitative comparisons in the revision to strengthen this point.
>
> 5. We have included a limitation section in Appendix E. We will move it to the main paper in the revision.
>
> **Minor points:** Thanks for mentioning these issues. We will improve the table and fix the typos in the revision.
>
> We hope the responses above have helped to resolve your concerns, and we remain available to answer any further questions. Thanks again for your insightful feedback!
>
> [1] Girdhar R, Singh M, Brown A, Duval Q, Azadi S, Rambhatla SS, Shah A, Yin X, Parikh D, Misra I. Emu video: Factorizing text-to-video generation by explicit image conditioning. arXiv preprint arXiv:2311.10709. 2023 Nov 17.

---

### Review · Reviewer_36k8 · 2024-05-01

**Summary Of Contributions:**

This paper extends text-to-image generation models, such as the Stable Diffusion series, to text-to-video generation. To achieve this, the authors first adapt intra-image cross-attention into a factorized spatial-temporal cross-attention. In the spatial attention stage, the first frame is attended with all other frames. In the temporal attention stage, tokens at the same position are attended to each other as well as to the tokens on the first frame within a 2D window. Furthermore, to enhance the temporal consistency of the generated video, the authors perform frequency analysis on the features of multiple video frames, and initialize the noisy latent images using the low-frequency component of the first frame. To prove the effectiveness of the proposed method, the authors conduct experiments quantitatively both on the UCF-101 and MSR-VTT datasets, and the proposed I2V-Bench. Ablation studies are also conducted to demonstrate the effectiveness of the proposed factorized spatial-temporal attention and low-frequency feature initialization.

**Audience:**

Yes

**Broader Impact Concerns:**

The potential broader impact has been sufficiently discussed in the paper.

**Claims And Evidence:**

Yes

**Requested Changes:**

- First, I would appreciate it if the authors could address the weaknesses, especially with a deeper analysis of the factors that limit the dynamics and motion magnitude of the generated video, both of which could potentially be impacted by the low-frequency initialization and the spatial-temporal attention.

- Another suggestion is to consider not sharing the same UNet for both the first image generation and the video generation. Could we keep the text-to-image generation fixed and only fine-tune the UNet for subsequent frames? This approach might allow us to use a relatively stronger fixed text-to-image generation network for visual quality, while having another UNet that leverages its full capacity on the dynamics and temporal consistency.

- Lastly, it would be beneficial to include a discussion with a recent paper that applies a diffusion process to components with different frequencies: https://openreview.net/pdf?id=OjDkC57x5sz

**Strengths And Weaknesses:**

Strengths:
- The authors clearly introduce the problems, which motivate the design of the paper.
- The frequency analysis opens a new direction for further research on improving consistency in an alternative way.
- The experiments are comprehensive, demonstrating not only the effectiveness of the proposed method but also its versatility in section 5.5.

Weaknesses:

- The major weakness of the proposed method is that although the low-frequency latent image initialization maintains temporal consistency, it potentially limits the content dynamics of the video.

  - Limited motion magnitude is mentioned in Appendix E, Limitation.2, which can be restricted by both the static low-frequency initialization and the limited window size K. A more detailed analysis would be beneficial, specifically examining how motion changes with the stop freqeucny in eq.5 and the size of K.

  - Another issue not addressed by the low-frequency frame initialization is the perspective view change with parallax. The low-frequency component of perspective change is still temporally varying; thus, initializing all other frames with the low-frequency component from the first frame can not effectively handle perspective changes. Therefore, besides 2D panning and zoomming, it would be also benificial to study perspective view change.

- The decoupled spatial-temporal attention mechanism needs further justification. It is understandable that the first frame needs to be highlighted in attention; however, the noisy latent images seem redundantly attended to the first frame both in spatial and temporal attention, which could further limit the dynamics of the generated video. It would be beneficial to see if the noisy images and the first frame could be equally attended during temporal attention, such as by using the same window size for both.

---

> ### Author Response · Authors · 2024-05-06
> **Response to Reviewer 36k8**
>
> We thank the reviewer for providing insightful reviews and feedback. We would like to respond to the weaknesses and requested changes in the following posts.
>
> **Responses to Weaknesses**
>
> > motion magnitude can be restricted by both the static low-frequency initialization and the limited window size K. A more detailed analysis would be beneficial
>
> We would like to clarify each of these two aspects separately. We will also include these discussions in the revision.
>
> 1. Low-frequency noise initialization (FrameInit): we did find that the motion magnitude can be restricted by FrameInit. Specifically, we observe that decreasing $\tau$ and increasing $D_0$ both lead to reduced motion magnitude. However, we also note that as the static first frame video is being perturbed by first adding $\tau$ steps of inference noise and then only preserving the low-frequency component based on the stop frequency $D_0$, we observe that this layout condition is not as strong as some other conditional signals, such as the DDIM inverted latents. The generated video will not always follow the layout condition and can still show diverse foreground/background motion as well as camera motion.
> 2. Temporal attention window size K: as mentioned in Section 3.4 of the paper, vanilla temporal attention is equivalent to K=1, while our method employs a larger K of 3 during training and inferencing. Therefore, our selected window size will include more first frame features during attention and will not restrict the motion magnitude compared to previous work. To further illustrate this point, we conduct additional experiments of training models with different K sizes on UCF-101 and evaluate the motion magnitude of 2048 generated videos using the “Dynamic Degree” metric from I2V-Bench. The results are shown in the table below:
>
>     | Window Size |  K=1 (Vanilla) |    K=3 (Ours) |  K=5  |  K=7  |
>     |----------------|:--------------:|:-------------:|:-----:|:-----:|
>     | Dynamic Degree |      68.46     |   **69.43**   | 64.11 | 68.01 |
> According to the table, our selected K=3 yields the highest motion magnitude and we did not find any correlation between the generated video’s dynamic degree and the window size K. This shows that the window-based temporal feature conditioning will not affect the motion magnitude of the generated videos.
> > perspective view change with parallax cannot be addressed by the low-frequency frame initialization
> >
>
> We appreciate the reviewer for pointing out this crucial problem for camera motion control. First, we would like to clarify that while the proposed FrameInit cannot directly support perspective view changes, as the low-frequency component of the static first frame video does not completely determine the camera motion in the output video (mentioned above), the synthesized video after applying FrameInit can still show diverse camera motion including perspective view changes (e.g. robot DJ case in Figure 10, best viewed with video file in supplementary materials). Furthermore, we also provide a simple adaptation of FrameInit to directly synthesize perspective view changes: given the input first frame, we apply image warping to create a sequence of perspective transformations. We then use the low-frequency component of this synthetic perspective view change to generate a new video. An example is shown in the videos below:
>
> Video 1: Synthetic perspective view change https://i.ibb.co/7SWchCp/synth-view-change.gif
>
> Video 2: Generated video https://i.ibb.co/WVV3JS4/generated-video.gif
>
> In this example, we select four key points in the first frame and determine their destination coordinates in the last frame. We then linearly interpolate the key point coordinates in the intermediate frames to create a synthetic camera motion containing perspective view changes (Video 1). By using the low-frequency component of this synthetic camera motion as the model input, we are able to generate videos that correctly reflect this perspective view change (Video 2). The FrameInit hyperparameters for this example are selected as $\tau=700$ and $D_0=0.5$ to enforce a stronger layout conditioning. We will update this example in the revision.

---

> > ### Author Response · Authors · 2024-05-06
> > **Response to Reviewer 36k8 (Cont'd)**
> >
> > > decoupled spatial-temporal attention mechanism needs further justification
> >
> > Our spatial and temporal first frame feature conditioning mechanisms are designed to tackle different problems during video generation: spatial conditioning ensures the video’s subject and background appearances are consistent with the provided first frame, and temporal conditioning enhances the temporal consistency of the generated video and reduces abrupt motion and flickering issues. As illustrated in Section 5.4, each component is necessary and we believe the first frame is not redundantly attended during temporal attention. Also, as mentioned above, attending to the extra first frame features during temporal attention will not restrict the dynamics of the generated videos.
> >
> > For the suggestion of letting each token attend to the same window in not only the first frame but also the noisy frames during temporal attention, we train our model using the default setting (K=3) on UCF-101 and compare it against the suggested method (termed as Tube Temporal Attention, as each token is attending to a spatiotemporal “tube” in the video). The results on 2048 videos are shown in the table below:
> >
> > |                               | FVD $\downarrow$ | IS $\uparrow$ | FID $\downarrow$ |
> > |-------------------------------|:----------------:|:--------------:|:----------------:|
> > | ConsistI2V (K=3)              |    **129.08**    |    **70.48**   |     **9.49**     |
> > | Tube Temporal Attention (K=3) |      168.48      |      70.04     |       10.66      |
> >
> > We can see that under the same window size, the suggested method did not outperform our method, and thus did not further enhance the generated video quality.
> >
> > **Other Requested Changes:**
> > > consider not sharing the same UNet for both the first image generation and the video generation.
> > >
> >
> > Our model in principle already follows this paradigm. We train a model to perform image(+
> > text)-to-video generation, meaning that the UNet receives 1) the first frame image and 2) the text prompt that describes the video, and then outputs a denoised version of the subsequent frames. This first frame image can be either a real-world image, or it can be generated by any off-the-shelf image generators that don’t need to share weights with the UNet in ConsistI2V. In particular, part of the qualitative results in Figures 5, 6, 7, 9 and 10 employ first frame images generated from image generators such as PixArt-$\alpha$ [1] and SDXL [2]. Therefore, we term our method as an image-to-video (I2V) generation approach or a two-stage text-to-video (T2V) generation method that leverages a pretrained image generator to generate the first frame in the first stage and animates this first frame to get the entire video in the second stage. We focused on introducing the I2V aspect of our model in the main paper, but we also included an additional quantitative analysis and compared our method against other T2V models in Appendix G.
> >
> > > beneficial to include a discussion with a recent paper https://openreview.net/pdf?id=OjDkC57x5sz
> > >
> >
> > Thank you very much for the suggestion. We will include this discussion in the revision.
> >
> >
> > We hope the above clarifications help address your concerns. We are also willing to resolve any further questions. Thanks again for the comprehensive review!
> >
> > [1] Chen J, Yu J, Ge C, Yao L, Xie E, Wu Y, Wang Z, Kwok J, Luo P, Lu H, Li Z. PixArt-$\alpha$: Fast Training of Diffusion Transformer for Photorealistic Text-to-Image Synthesis. arXiv preprint arXiv:2310.00426. 2023 Sep 30.
> >
> > [2] Podell D, English Z, Lacey K, Blattmann A, Dockhorn T, Müller J, Penna J, Rombach R. Sdxl: Improving latent diffusion models for high-resolution image synthesis. arXiv preprint arXiv:2307.01952. 2023 Jul 4.

---

### Author Response · Authors · 2024-05-13
**Manuscript Revision**

We thank all the reviewers for their detailed suggestions and feedback. We have incorporated the requested changes in the revised manuscript. A detailed change log since the last submission is shown as follows:

(All the changes in the revision are highlighted in blue)
1. Section 3.4: add discussion on window size K (requested by Reviewer hVB1)
2. Section 3.5: add discussion with Blurring Diffusion Model (requested by Reviewer 36k8)
3. Table 1 & Table 2: change text color and highlight (requested by Reviewer hVB1)
4. Figure 5: change example with less motion (requested by Reviewer ashH)
5. Figure 6: add two more ablation study example (requested by Reviewer hVB1)
6. Table 3: include inference runtime statistics (requested by Reviewer hVB1)
7. Section 5.4: add discussion on FrameInit hyperparameters; add discussion on runtime efficiency (requested by Reviewer 36k8, ashH, hVB1)
8. Figure 7: add a new figure for FrameInit hyperparameter comparison (requested by Reviewer 36k8, ashH)
9. Section 5.5 & Figure 8: add discussion & visualization for perspective view change (requested by Reviewer 36k8)
10. Section 6: move the Limitation section to main paper (requested by Reviewer hVB1)
11. Appendix C: add more descriptions for I2V-Bench metrics (requested by Reviewer ashH)

Please don't hesitate to reach out if you have any further questions regarding the manuscript and we are more than happy to assist.

---

### Decision · Action_Editor_y4PD · 2024-06-22

**Recommendation:** Accept as is

**Comment:**

This paper addresses Image-to-Video (I2V) generation, specifically the problem of generating a video conditioned on a text prompt and an initial image. The paper's goal is to mitigate appearance and motion inconsistency in the generated video. The authors introduce the ConsistI2V diffusion model, which contributes to achieving this goal in two ways: 1) by introducing a spatio-temporal cross-attention conditioning over the first frame in the diffusion model, and 2) by designing a noise initialization that uses the low-frequency band from the first frame (FrameInit). The authors also propose a comprehensive evaluation benchmark (I2V-Bench) and demonstrate the importance of the proposed contributions through ablation studies.

The paper initially received positive feedback, with reviewers appreciating the simplicity of the approach, the relevance of the contributions, and the solidity of the experiments. They also raised concerns and requested further justifications for the proposed approach, a discussion on the computation's overhead induced by ConsistI2V, and clarifications on experiments. The rebuttal did a good job in addressing the most important reviewers' concerns. Although there remained some limitations regarding novelty, the reviewers considered that the claims are sufficiently validated by evidence, and there was a consensus to accept the paper.

The AE carefully reviewed the submission and discussions. The AE considers that the approach is a timely and meaningful adaptation of Image-to-Video (I2V) generation to improve generation consistency. The practical benefits of the two introduced contributions are clearly validated in the experiments. The I2V benchmark introduced by the authors is also valuable for the community. Therefore, the AE recommends acceptance.

**Audience:**

The paper addresses Image-to-video (I2V) generation, a major topic in the current context of machine learning, which will be of interest to a wide TMLR audience.

**Claims And Evidence:**

The claims are convincingly supported by evidence.